# Finding Differentially Private Second Order Stationary Points in Stochastic Minimax Optimization

Difei Xu [* 1 2]  Youming Tao [* 2 3]  Meng Ding [2 4]  Chenglin Fan [2 5]  Di Wang [1 2]

## Abstract

We provide the first study of the problem of finding differentially private (DP) second-order stationary points (SOSP) in stochastic (non-convex) minimax optimization. Existing literature either focuses only on first-order stationary points for minimax problems or on SOSP for classical stochastic minimization problems. This work provides, for the first time, a unified and detailed treatment of both empirical and population risks. Specifically, we propose a purely first-order method that combines a nested gradient descent–ascent scheme with SPIDER-style variance reduction and Gaussian perturbations to ensure privacy. A key technical device is a block-wise ($q$-period) analysis that controls the accumulation of stochastic variance and privacy noise without summing over the full iteration horizon, yielding a unified treatment of both empirical-risk and population formulations. Under standard smoothness, Hessian-Lipschitzness, and strong concavity assumptions, we establish high-probability guarantees for reaching an $(\alpha, \sqrt{\rho_\Phi \alpha})$-approximate second-order stationary point with $\alpha = \mathcal{O}((\frac{\sqrt{d}}{n\varepsilon})^{2/3})$ for empirical risk objectives and $\mathcal{O}(\frac{1}{n^{1/3}} + (\frac{\sqrt{d}}{n\varepsilon})^{1/2})$ for population objectives, where $n$ is the number of samples, matching the best known rates for private first-order stationarity, $d$ is the dimension and $\varepsilon$ is the privacy parameter.

*Equal contribution [1]King Abdullah University of Science and Technology [2]Provable Responsible AI and Data Analytics (PRADA) Lab [3]Technische Universität Berlin [4]State University of New York at Buffalo [5]Seoul National University. Correspondence to: Di Wang <di.wang@kaust.edu.sa>.

*Proceedings of the 43rd International Conference on Machine Learning*, Seoul, South Korea. PMLR 306, 2026. Copyright 2026 by the author(s).

## 1. Introduction

Stochastic optimization plays a central role in modern machine learning. Among its variants, minimax optimization—an important instance of hierarchical optimization—has found broad applications across diverse machine learning problems, including Generative Adversarial Networks (Goodfellow et al., 2014), adversarial training (Madry et al., 2017; Fu et al., 2025; Fu & Wang, 2023), multi-agent reinforcement learning (Wai et al., 2018), as well as meta-learning and hyperparameter optimization (Ren et al., 2025). In recent years, extensive research efforts have been devoted to the theoretical and algorithmic study of minimax optimization in various machine learning settings. A wide range of deterministic and stochastic methods have been proposed, accompanied by both asymptotic and non-asymptotic convergence analyses, such as Gradient Descent Ascent (GDA) (Du & Hu, 2019; Nemirovski, 2004) and Stochastic Gradient Descent Ascent (SGDA). Some approaches employ a single-loop update scheme (Heusel et al., 2017), while others adopt a nested-loop structure that updates the inner variable $y$ more frequently in order to obtain a more accurate approximation of the maximizer $y^*(x)$ (Jin et al., 2020).

Despite these advances, the majority of existing work focuses on convergence to first-order stationary points. In nonconvex settings, however, such a notion of optimality is often insufficient, as a first-order stationary point may correspond to a local minimum, a saddle point, or even a local maximum. This limitation has motivated growing interest in second-order stationary points, which characterize local minima and thus provide a stronger notion of solution quality in nonconvex optimization. Since finding a global minimum in general nonconvex problems is typically NP-hard (Hillar & Lim, 2013), practical algorithms often aim to identify local minima instead. Moreover, in certain machine learning applications—such as tensor decomposition (Ge et al., 2015) and matrix sensing (Bhojanapalli et al., 2016)—all local minima are global minima, rendering the pursuit of SOSP particularly critical.

Because training data are typically sensitive, differential privacy (DP) (Dwork et al., 2006) has become a standard requirement for learning with principled privacy guarantees. While hundreds of studies on differential privacy (DP) have

*Table 1.* Summary of prior work on DP both Minimax and Minimal problems compared with our proposed approach. Here, $d$ denotes the dimensionality of the parameter variable and $n$ the sample size. When applying minimax algorithms to SOSP problems.

| Methods | Problem | Objective function | Utility bound | SOSP |
|---|---|---|---|---|
| DP SPIDER (Arora et al., 2023) | Minimal | Empirical | $\tilde{\mathcal{O}}((\frac{\sqrt{d}}{n\varepsilon})^{2/3})$ | ✗ |
| Stochastic SPIDER (Liu et al., 2023) | Minimal | Empirical | $\mathcal{O}((\frac{\sqrt{d}}{n\varepsilon})^{2/3})$ | ✓ |
| DP RGDA (This work) | Minimax | Empirical | $\mathcal{O}((\frac{\sqrt{d}}{n\varepsilon})^{2/3})$ | ✓ |
| DP SGDA (Yang et al., 2022) | Minimax | Population | $\mathcal{O}(\frac{1}{\sqrt{n}} + \frac{\sqrt{d}}{n\varepsilon})$ | ✗ |
| Ada-DP-SPIDER (Tao et al., 2025) | Minimal | Population | $\mathcal{O}(\frac{1}{n^{1/3}} + (\frac{\sqrt{d}}{n\varepsilon})^{2/5})$ | ✓ |
| Stochastic SPIDER with Escaping Procedure (Liu & Talwar, 2024) | Minimal | Population | $\mathcal{O}(\frac{1}{n^{1/3}} + (\frac{\sqrt{d}}{n\varepsilon})^{1/2})$ | ✓ |
| DP RGDA (This work) | Minimax | Population | $\mathcal{O}(\frac{1}{n^{1/3}} + (\frac{\sqrt{d}}{n\varepsilon})^{1/2})$ | ✓ |

been developed for empirical risk minimization (ERM), i.e., single-level minimization of an average loss over the last decade (Bassily et al., 2014; 2019; Xu et al., 2025; Tao et al., 2022; Wang et al., 2020a; 2017; Wang & Xu, 2019; Hu et al., 2022; Su et al., 2024; Xue et al., 2021), recent attention has shifted toward privately finding second-order stationary points (DP-SOSP) (Wang et al., 2019; 2020b). On the other hand, while several recent studies have explored DP minimax optimization, all of them focus on either convex-concave settings (Rafique et al., 2022; Zhou & Bassily, 2024) or first-order stationary points (Zhang et al., 2025b) (we defer a detailed summary of these minimization-focused rates to Table 1). Thus, to the best of our knowledge, no existing work addresses finding DP-SOSP for minimax optimization, which motivates this study.

Compared to classical DP-SOSP for stochastic minimization, establishing DP-SOSP guarantees for nonconvex–strongly-concave minimax objectives introduces new technical obstacles: the target is the value function $\Phi(x) = \max_{y \in \mathcal{Y}} f(x, y)$, whose gradient depends on an inner maximizer $y^\star(x)$ that must be tracked approximately, and the privacy noise injected into both ascent and descent updates propagates through the nested dynamics, making it nontrivial to preserve the accuracy required for second-order stationarity of $\Phi$. In this work, we provide the first algorithmic framework that targets DP-SOSP for stochastic minimax optimization: our method combines SPIDER variance reduction for the coupled $(x, y)$ updates with a perturb-and-monitor saddle-escape mechanism based on iterate displacement, avoiding explicit Hessian computations of $\Phi$. As a byproduct, when the minimax structure degenerates to a pure minimization problem, our framework recovers the state-of-the-art DP-SOSP guarantees for ERM and matches the best-known population rate of Liu & Talwar (2024).

Our contribution can be summarized as follows:

1. **A Generic Framework for Stochastic nonconvex Minimax Optimization** We are the first to propose a generic framework that uses the SPIDER variance reduction technique to improve the bound for the differentially private gradient descent ascent method for nonconvex-strongly convex minimax optimization problems. We add an inner loop for the update of variable $y$ so that we can depict the convergence more conveniently. By introducing the inner iteration number $K$, a constant, our algorithm can achieve state-of-the-art results without the Hessian matrix, purely using gradient methods.

2. **The Best Results as Finding the First-order Stationary Points** We prove that we achieve $\alpha$ second order stationary points with $\alpha = \tilde{O}((\frac{\sqrt{d}}{n\varepsilon})^{2/3})$ for empirical loss and $\alpha = \tilde{O}(\frac{1}{n^{1/3}} + (\frac{\sqrt{d}}{n\varepsilon})^{1/2})$ for population loss functions, which matches the best results of finding the first-order stationary point in the same problem.

3. We implement our algorithms on synthetic matrix sensing problems, demonstrating performance comparable to previous methods that only achieve first-order stationary points.

## 2. Preliminaries

### 2.1. Differential Privacy

**Definition 1** (Differential Privacy (Dwork et al., 2006))**.** Given a data universe $\mathcal{X}$, we say that two datasets $S, S' \subseteq \mathcal{X}$ are neighbors if they differ by only one entry, which is denoted as $S \sim S'$. A randomized algorithm $\mathcal{A}$ is $(\varepsilon, \delta)$-differentially private (DP) if for all neighboring datasets $S, S'$ and for all events $E$ in the output space of $\mathcal{A}$, the following holds

$$\mathbb{P}(\mathcal{A}(S) \in E) \leqslant e^\varepsilon \mathbb{P}(\mathcal{A}(S') \in E) + \delta.$$

If $\delta = 0$, we call algorithm $\mathcal{A}$ $\varepsilon$-DP.

**Definition 2.** For a given function $q : \mathcal{Z} \to \mathbb{R}^d$, we say $q$ has $\Delta_2(q)$ $\ell_2$-sensitivity if for any neighboring datasets $D, D'$ we have $\|q(D) - q(D')\| \leqslant \Delta_2(q)$.

**Definition 3.** For a given function $q : \mathcal{Z} \to \mathbb{R}^d$, the Gaussian mechanism is defined as $q(D) + \xi$ where $\xi \sim \mathcal{N}\left(0, \frac{\Delta_2^2(q)\log(1.25/\delta)}{\varepsilon^2}\mathbf{I}_d\right)$. Gaussian mechanism preserves $(\varepsilon, \delta)$-DP for $0 < \varepsilon, \delta < 1$.

### 2.2. Second Order Stationary Points for Minimax Optimization

We will present the notations that will be useful for the statement of the problem setting and clarify the statement of the results. We assume that the function $f$ is twice differentiable. Notation $\tilde{\mathcal{O}}$ means the complexity after hiding logarithmic terms.

In this paper, we study the following stochastic minimax optimization problem,

$$\min_{x \in \mathbb{R}^{d_1}} \max_{y \in \mathcal{Y} \subseteq \mathbb{R}^{d_2}} f(x, y) = \mathbb{E}[F(x, y; \xi)], \qquad (1)$$

where the function $f$ is smooth and possibly non-convex in variable $x$ while being smooth and strongly-concave in variable $y$, which is essential to make SOSP well defined. In addition, $\xi$ and $\xi_i$, which will be used below, are samples drawn from the data distribution $\mathcal{D}$.

Similarly, given a training dataset $S = \{\xi_i\}_{i=1}^n$, for the empirical loss, we have

$$\min_{x \in \mathbb{R}^{d_1}} \max_{y \in \mathcal{Y} \subseteq \mathbb{R}^{d_2}} f_S(x, y) = \frac{1}{n}\sum_{i=1}^{n}[F(x, y; \xi_i)], \qquad (2)$$

where $\mathcal{Y}$ is a convex domain (not required to be compact). The loss function $f(x, y)$ is smooth and possibly nonconvex w.r.t. $x$, and is smooth and strongly-concave w.r.t. $y$.

To simplify the discussion below of our analysis, we need the following definition: we define $\Phi(x)$ as:

$$\Phi(x) := \max_{y \in \mathcal{Y} \subseteq \mathbb{R}^{d_2}} f(x, y).$$

and for the empirical loss $\Phi_S(x) := \max_{y \in \mathcal{Y} \subseteq \mathbb{R}^{d_2}} f_S(x, y)$.

Since the empirical form can be taken as a form of the uniform distribution over the dataset, we use $f$ and $\Phi$ for the general discussion, which should not cause any ambiguity.

**Definition 4.** (Xian et al., 2025) A point $x$ is called $\alpha$-SOSP if it satisfies the following expression: $\|\nabla\Phi(x)\| \leqslant \alpha$ and $\lambda_{\min}(\nabla^2\Phi(x)) \geqslant -\alpha_H$, where $\lambda_{\min}$ denotes the smallest eigenvalue and $\alpha_H = \sqrt{\rho_\Phi \alpha}$ and $\rho_\Phi$ is the Lipschitz constant of $\nabla^2\Phi(x)$.

---

**Algorithm 1** Clipping $(x, C)$

**Require:** $x$ and clipping threshold $C > 0$.
1: $\hat{x} = \min\left\{\frac{C}{\|x\|_2}, 1\right\} x$
**Ensure:** $\hat{x}$.

---

Having defined SOSP, we introduce the following assumptions on Lipschitz continuity of first and second order derivatives.

**Assumption 1.** The function $F(x, y)$ is $M$-Lipschitz over each coordinate.

Similarly, we need the Lipschitzness of the gradient of the loss functions.

**Assumption 2.** The gradients of component functions $F(x, y; \xi)$ are $L$-Lipschitz continuous, i.e., there exists a constant $L$ such that

$$\|\nabla F(z; \xi) - \nabla F(z'; \xi)\| \leqslant L\|z - z'\|, \qquad (3)$$

for any $z = (x, y)$ and $z' = (x', y')$.

The two assumptions above are standard for optimization in both differentially private settings (Tao et al., 2025; Liu & Talwar, 2024; Arora et al., 2023; Zhang et al., 2025a; Huai et al., 2020; Su et al., 2023) and non-private settings (Xian et al., 2025; Luo et al., 2022). A minor distinction is that we assume the Lipschitz property holds coordinatewise, rather than only with respect to the inner and outer variables.

**Assumption 3.** The second order derivatives $\nabla_x^2 F(x, y)$, $\nabla_{xy}^2 F(x, y)$, $\nabla_y^2 F(x, y)$ are $\rho$-Lipschitz continuous.

**Assumption 4.** $g(x, y) := -F(x, y)$ is $\mu$-strongly convex with respect to $y$, i.e. there exists a constant $\mu$ such that

$$g(x, y) + \langle \nabla_y g(x, y), y' - y \rangle + \frac{\mu}{2}\|y' - y\|^2 \leqslant g(x, y')$$

for any $x, y$ and $y'$.

With Assumption 4, the objective function $\Phi(x)$ is also differentiable and the gradient is formulated as follows:

$$\nabla\Phi(x) = \nabla_x f(x, y^*(x)) +$$
$$\nabla_{xy}^2 f(x, y^*(x))[\nabla_y^2(-f)(x, y^*(x))]^{-1}\nabla_y f(x, y^*(x)).$$

In minimax optimization, since we always have $\nabla_y f(x, y^*(x)) = 0$, the expression of $\nabla\Phi(x)$ can be simplified by

$$\nabla\Phi(x) = \nabla_x f(x, y^*(x)). \qquad (4)$$

**Definition 5.** We define the condition number $\kappa$ as $\kappa = \frac{L}{\mu}$, where $L$ is the Lipschitz constant.

Additionally, we need one extra assumption to guarantee the convergence of one term in our proof:

**Assumption 5.** We assume that each component function $F(x, y; \xi)$ satisfies bounded variance, *i.e.*,

$$\|\nabla F(x, y; \xi) - \nabla f(x, y)\| \leqslant \sigma. \qquad (5)$$

---

**Algorithm 2** Differentially Private Recursive Gradient Descent Ascent (DP RGDA)

---

**Require:** initial value $x_0, y_0$, stepsize $\eta$ and $\eta_H$, perturbation radius $r$, escaping phase threshold $t_{thres}$, average movement $\bar{D}$, maximum iteration $T$.

1: Set $escape = \textbf{False}, s = 0, esc = 0$.
2: **for** $t = 0, 1, \ldots, T - 1$ **do**
3:     Update $y_{t+1}, v_t, u_t$ from Algorithm 3.
4:     **if** $escape = \textbf{False}$ **then**
5:         **if** $\|v_t\| \geqslant \alpha$ **then**
6:             Update $x_{t+1} = x_t - (\eta/\|v_t\|)v_t$.
7:         **else**
8:             Let $m_s = t, s = s + 1, escape = \textbf{True}, esc = 0$.
9:             Draw perturbation $\xi \sim B_0(r)$ and update $x_{t+1} = x_t + \xi$.
10:         **end if**
11:     **else**
12:         Compute $D = \sum_{j=m_s+1}^{t} \eta_H^2 \|v_j\|^2$.
13:         **if** $D > (t - m_s)\bar{D}$ **then**
14:             Set $\eta_t$ s.t. $\sum_{j=m_s+1}^{t} \eta_t^2 \|v_j\|^2 = (t - m_s)\bar{D}$.
15:             Update $x_{t+1} = x_t - \eta_t v_t$. Set $escape = \textbf{False}$.
16:         **else**
17:             Set $\eta_t = \eta_H$.
18:             Update $x_{t+1} = x_t - \eta_t v_t, esc = esc + 1$.
19:             **Return** $x_{m_s}$ if $esc = t_{thres}$.
20:         **end if**
21:     **end if**
22: **end for**
**Ensure:** $x_{m_s}$

---

## 3. Challenges

Our objective in the nonconvex–(strongly) concave setting is the minimax value function $\Phi(x) := \max_{y \in \mathcal{Y}} f(x, y)$, defined either for the population loss or for the empirical risk. We seek an $\alpha$-second-order stationary point (SOSP) of $\Phi$, i.e., a point with small $\|\nabla\Phi(x)\|$ and nearly nonnegative curvature. This is a fundamentally different target from stationarity of the saddle objective $f$: even when $f(x, \cdot)$ is $\mu$-strongly concave, a stationary pair $(x, y)$ for simultaneous descent–ascent dynamics does not certify that $x$ avoids strict saddles of $\Phi$ unless $y$ tracks the maximizer $y^\star(x)$ accurately.

SGDA is the canonical baseline for minimax learning, and DP-SGDA (Rafique et al., 2022) is its natural private variant. However, the standard SGDA theory mainly certifies first-order criteria, which do not rule out strict saddles of

$\Phi$ and therefore do not imply convergence to local minimizers of $\Phi$. Moreover, SGDA has no built-in mechanism to enforce second-order behavior. Augmenting DP-SGDA with common saddle-escape or model-selection primitives is problematic here: function-decrease tests for $\Phi$ are not directly implementable since $\Phi$ is implicit, and privately estimating Hessian information of $\Phi$ (e.g., $\lambda_{\min}(\nabla^2\Phi(x))$) is high-sensitivity and expensive. Thus, DP-SGDA can serve as a heuristic baseline, but it does not directly yield a DP-SOSP guarantee for the value function.

Existing DP algorithms for SOSP in single-level minimization operate on an objective with direct gradient access and pay privacy only for those gradients. In minimax problems, the relevant direction is implicit: $\nabla\Phi(x) = \nabla_x f(x, y^\star(x))$. Any implementable update must therefore control, simultaneously, (i) the bias from approximating $y^\star(x)$, (ii) sampling noise (for population objectives), and (iii) DP noise from privatizing every gradient access. This interaction is amplified by the nested access pattern: one outer step typically consumes multiple inner gradient calls, so naive noise injection can accumulate rapidly. In addition, DP composition must be allocated across heterogeneous oracle types: infrequent large-batch refresh queries and frequent small-batch incremental updates have different sensitivity/composition behavior, forcing batch sizes and refresh frequencies to be chosen jointly. Finally, excluding strict saddles of $\Phi$ using standard DP tests (e.g., repeated private Hessian-eigenvalue checks, or private selection over iterates) is typically prohibitively costly.

We combine four ingredients that are tailored to these bottlenecks. (1) The strong concavity of the inner problem allows us to convert an inner-loop stationarity surrogate into control of $\|y - y^\star(x)\|$, and hence into a bound on the bias $\|\nabla_x f(x, y) - \nabla\Phi(x)\|$. (2) We use a SPIDER recursion (Arora et al., 2023) to track the relevant gradients across iterations, reducing the number of expensive refresh queries and limiting DP noise accumulation. (3) The analysis is organized into short periods aligned with the refresh schedule, which is essential in the population setting where sampling noise persists and full-horizon telescoping is unavailable. (4) We replace DP saddle point certification and private model selection with a perturb-and-monitor criterion based on iterate displacement, which is computable from privatized gradients and therefore incurs no additional privacy loss. Together, these choices yield a first-order DP method that targets SOSP of the value function $\Phi$ without explicit Hessian computations.

## 4. Main Theory

This section presents the main theoretical guarantees of Algorithm 2 and 3 for privately computing approximate second-order stationary points of the minimax loss func-

---

**Algorithm 3** Updater of Inner Loop

---

**Require:** status $x_t, x_{t-1}, y_t, v_{t-1}, u_{t-1}$ and $t$

**Require:** stepsize $\lambda$, inner loop size $K$, batchsize $S_1$ and $S_2$, period $q$.

1: $\sigma_{\omega_t} = \sigma_{\tau_t} = \frac{C_v \sqrt{\log(1/\delta)}}{\varepsilon} \max\{\frac{1}{S_1}, \frac{\sqrt{T}}{\sqrt{q}n}\}$, where $c$ is a universal constant.

2: $\sigma_{\zeta_t} = \sigma_{\chi_t} = \frac{C_u \sqrt{\log(1/\delta)}}{\varepsilon} \cdot \max\{\frac{1}{S_2}, \frac{\sqrt{T}}{n}\}$, where we define $w_t = (x_y, y_t)$ for simplicity.

3: Set $x_{t,-1} = x_{t-1}$, $x_{t,k} = x_t$ when $k \geqslant 0$, $y_{t,-1} = y_{t,0} = y_t$.

4: **if** $\mathrm{mod}(t, q) = 0$ **then**

5:     Draw $S_1$ samples $\{\xi_1, \ldots, \xi_{S_1}\}$

6:     Compute:

7:     $v_{t,-1} = \omega_t + \mathbf{Clip}(\frac{1}{S_1} \sum_{i=1}^{S_1} \nabla_x F(x_t, y_t; \xi_i), C_v)$,

8:     $u_{t,-1} = \tau_t + \mathbf{Clip}(\frac{1}{S_1} \sum_{i=1}^{S_1} \nabla_y F(x_t, y_t; \xi_i), C_v)$.

9: **else**

10:     Let $v_{t,-1} = v_{t-1}$, $u_{t,-1} = u_{t-1}$.

11: **end if**

12: **for** $k = 0$ to $K - 1$ **do**

13:     Draw $S_2$ samples $\{\xi_1, \ldots, \xi_{S_2}\}$

14:     Compute $\quad v_{t,k} \quad = \quad v_{t,k-1} \quad + \quad \zeta_{t,k} \quad +$ $\mathbf{Clip}(\frac{1}{S_2} \sum_{i=1}^{S_2} (\nabla_x F(x_{t,k}, y_{t,k}; \xi_i) \quad - \nabla_x F(x_{t,k-1}, y_{t,k-1}; \xi_i)), C_u)$

15:     Compute $\quad u_{t,k} \quad = \quad u_{t,k-1} \quad + \quad \chi_{t,k} \quad +$ $\mathbf{Clip}(\frac{1}{S_2} \sum_{i=1}^{S_2} (\nabla_y F(x_{t,k}, y_{t,k}; \xi_i) \quad - \nabla_y F(x_{t,k-1}, y_{t,k-1}; \xi_i)), C_u)$

16:     $y_{t,k+1} = \Pi_{\mathcal{Y}}(y_{t,k} + \lambda u_{t,k})$.

17: **end for**

18: Select $s_t = \arg\min_k \|\tilde{G}_\lambda(y_{t,k})\|$. Let $y_{t+1} = y_{t,s_t}$, $v_t = v_{t,s_t}$, $u_t = u_{t,s_t}$.

**Ensure:** $y_{t+1}, v_t, u_t$.

---

tion. Throughout, we focus on functions (1) and (2), corresponding to the population loss and the empirical risk, respectively.

Algorithm 2 is an outer-loop driver that calls Algorithm 3 as a data-accessing subroutine. At each outer iteration $t$, Algorithm 3 performs $K$ projected ascent steps on $y$ at the fixed anchor $x_t$, and maintains two recursive sequences:

$$u_{t,k} \approx \nabla_y f(x_t, y_{t,k}), \qquad v_{t,k} \approx \nabla_x f(x_t, y_{t,k}),$$

with an analogous interpretation for $f$ in the population case. The recursion is SPIDER: every $q$ outer iterations the estimators are refreshed using a batch of size $S_1$, and otherwise updated using a smaller batch $S_2$ and gradient differences. Gaussian perturbations are injected at each refresh and each recursive update to ensure differential privacy.

The only operations that access the dataset are the noisy gradient computations inside Algorithm 3. Consequently, once the sequences $\{u_{t,k}, v_{t,k}\}$ are made differentially pri-

vate, all subsequent operations in Algorithm 2 (including the updates of $x_t$, the selection of $y_{t+1}$, and the perturbations used for saddle escape) are post-processing and do not incur any additional privacy loss.

Algorithm 2 alternates between two modes: (i) a *descent phase*, activated when the estimated value-gradient magnitude is large, $\|v_t\| \geqslant \alpha$, where we take a normalized step of fixed length $\eta$ along $-v_t$; and (ii) an *escape phase*, activated when $\|v_t\| < \alpha$, where we apply a random perturbation of radius $r$ and then take $t_{\mathrm{thres}}$ steps with step size $\eta_H$. The escape phase is monitored by the movement statistic $D_t = \sum_{j=m_s+1}^t \eta_H^2 \|v_j\|^2$, and terminates early if the average squared movement exceeds the threshold $\bar{D}$. If the escape phase does not terminate early, Algorithm 2 outputs the anchor point $x_{m_s}$.

### 4.1. Key Idea and Proof Strategy

The central difficulty in differentially private minimax optimization is that (i) the value-gradient $\nabla\Phi(x) = \nabla_x f(x, y^\star(x))$ is not directly available, and (ii) naive recursive estimators accumulate privacy noise over a long horizon, which leads to vacuous control of the optimization error. Our approach addresses these two issues through three coupled design choices.

**(i) Enforce a value-function viewpoint by tracking $y^\star(x)$.** Algorithm 3 includes an inner loop of $K$ projected ascent updates on $y$ at each outer iterate $x_t$. The iterate $y_{t+1}$ is selected using the projected gradient mapping criterion, which is a standard stationarity surrogate for constrained maximization. Under $\mu$-strong concavity and definition of $G_\lambda$ as in Appendix B.2, this yields a quantitative bound of the form

$$\|y_{t+1} - y^\star(x_t)\| \lesssim \|G_\lambda(x_t, y_{t+1})\|,$$

which in turn converts control of the inner-loop stationarity into control of the bias term $\|\nabla_x f(x_t, y_{t+1}) - \nabla\Phi(x_t)\|$.

**(ii) Control the recursive estimator locally (period-wise) rather than globally.** The estimators are refreshed every $q$ outer iterations. Instead of requiring uniform control over all $T$ steps, we analyze the estimator error within each period and restart the argument at the next refresh. This is the point at which our analysis diverges from proofs that rely on telescoping bounds across the entire horizon: a period-wise control prevents privacy noise from accumulating unavoidably in the estimator deviation, and yields the rates stated in Theorems 1 and 2.

**(iii) Avoid private model selection via a movement-based escape test.** A common obstacle for DP nonconvex optimization is that selecting the "best" iterate may require additional private evaluation (e.g., AboveThreshold)(Liu

et al., 2023). Algorithm 2 avoids this overhead by using the movement statistic $D_t$ during the escape phase. The test is computable from privatized gradients and iterates only. Large movement certifies that the algorithm has escaped a region of negative curvature, while persistently small movement implies that the Hessian at the anchor point is nearly positive semidefinite. This mechanism yields approximate second-order stationarity without an extra DP selection step.

## 4.2. SOSP with Empirical Loss

In the ERM setting, the objective is a finite sum $f(x, y) = \frac{1}{n} \sum_{i=1}^{n} F(x, y; \xi_i)$ over a fixed dataset $S = \{\xi_i\}_{i=1}^{n}$. We instantiate Algorithm 3 with a full refresh batch $S_1 = n$ and a single inner update per outer step (i.e., $K$ is treated as a fixed constant; in particular, the proofs in the appendix specialize to $K = 1$). In this regime, the only randomness in the gradient oracle arises from the injected Gaussian noise, and the resulting utility bound is driven by the privacy term.

We first quantify how far the recursive estimator can drift from the fresh gradient estimate produced at refresh steps. Let

$$w_t := (x_t, y_t),$$
$$\nabla_t := \left( \frac{1}{S_1} \sum_{i \in S_1} \nabla_x F(w_t; \xi_i) + \omega_t \atop -\frac{1}{S_1} \sum_{i \in S_1} \nabla_y F(w_t; \xi_i) - \tau_t \right)^T,$$

and define the recursive estimator $\Delta_t := (v_t, -u_t)$. The following bound is a standard consequence of the SPIDER recursion (Proposition 1 in (Fang et al., 2018)).

**Lemma 1.** *Consider Algorithm 3, and for any $t \in \{0, ..., T\}$ let $t_0 = \left\lfloor \frac{t}{q} \right\rfloor q$. If each $\nabla_t$ computed defined above is an unbiased estimate of $\nabla F(w_t; S)$ satisfying with probability $1 - \delta_1$,*

$$\|\nabla_{t_0} - \nabla F(w_{t_0}; S)\|^2 \leq B_1^2 \log(1/\delta_1),$$

*and each $\Delta_t$ is an unbiased estimate of the gradient variation satisfying*

$$\|\Delta_t - [\nabla F(w_t; S) - \nabla F(w_{t-1}; S)]\|^2$$
$$\leqslant B_2^2 \|w_t - w_{t-1}\|^2 \log(1/\delta_1).$$

*Then for any $t \geqslant t_0 + 1$, the iterates of Algorithm 3 satisfy*

$$\|\nabla_t - \nabla F(w_t)\|^2$$
$$\leqslant \log(1/\delta_1)(B_2^2 \sum_{k=t_0+1}^{t} \|w_k - w_{k-1}\|^2 + B_1^2).$$

Lemma 1 is used to control the optimization error induced by the recursive updates and by the privacy noise. Combined with the smoothness of $\Phi$, it yields a descent inequality in

the descent phase (when $\|v_t\| \geqslant \alpha$), and it ensures that upon entering the escape phase (when $\|v_t\| < \alpha$), the true gradient $\|\nabla\Phi(x_t)\|$ is also small up to the same order. The escape phase then follows the standard perturbed-gradient paradigm: if the Hessian at the anchor has a sufficiently negative eigenvalue, the perturbation causes the iterates to travel a nontrivial distance with constant probability, which forces an early exit from the escape phase; otherwise, failure to travel implies near-positive semidefinite curvature at the anchor point.

**Theorem 1.** *Assume Assumptions 1–4 hold. For given privacy parameters $(\varepsilon, \delta)$ and failure probabilities $\delta_1, \delta_2 \in (0, 1)$. Run Algorithm 2 with $S_1 = n$, $S_2 \geqslant \max\{(\frac{Mn\varepsilon}{\sqrt{F_0 L d \log(1/\delta)}})^{2/3}, \frac{(Mnd\log(1/\delta))^{1/3}}{(LF_0)^{1/6}}\}$, where $F_0 = f_S(\mathbf{0}) - \min_{x,y \in \mathbb{R}^d}\{f_S(x, y)\}$, $L_\Phi$, and $\rho_\Phi$ are the gradient and Hessian Lipschitz constants of the value function. Define $B_0(r) := \xi \in \mathbb{R}^{d_1} : |\xi|2 \leq r$. Besides, esc means the escape steps and escape means the escape phase in our algorithm. Choose the remaining parameters as in Appendix B.1, Algorithm 2 is $(\varepsilon, \delta)$-DP and outputs a point $x_{\text{out}}$ such that, with probability at least $1 - \delta_1 - \delta_2$,*

$$\|\nabla\Phi_S(x_{\text{out}})\| \leq \alpha, \lambda_{\min}(\nabla^2\Phi_S(x_{\text{out}})) \geq -\sqrt{\rho_\Phi \alpha},$$

*with $\alpha = \widetilde{\mathcal{O}}(\bar{\epsilon}^{2/3}) = \widetilde{\mathcal{O}}\left(\left(\frac{\sqrt{d\log(1/\delta)}}{n\varepsilon}\right)^{2/3}\right).$*

Theorem 1 states that, for ERM, the stationarity accuracy is dominated by the privacy term $\bar{\epsilon} = \frac{\sqrt{d\log(1/\delta)}}{n\varepsilon}$, and the algorithm achieves a $\widetilde{\mathcal{O}}(\bar{\epsilon}^{2/3})$-approximate SOSP using only first-order access through privatized gradients.

*Remark 1.* Motivated by the privacy requirement in learning, a growing line of work studies how to compute approximate second-order stationary points (SOSP) under differential privacy constraints; see, e.g., (Liu et al., 2023) for DP-SOSP algorithms in non-convex *minimization*/ERM. Theorem 1 shows that, in the empirical-risk minimax setting, our method attains $\alpha = \tilde{O}((\sqrt{d}/(n\varepsilon))^{2/3})$ (up to logarithmic factors), which matches the best-known ERM DP-SOSP scaling under comparable smoothness assumptions.

## 4.3. SOSP with Population Loss

We now consider the population objective $f(x, y) = \mathbb{E}_\xi[F(x, y; \xi)]$. Compared with ERM, two additional issues arise: (i) the gradient oracle has intrinsic stochastic variance (Assumption 5), and (ii) privacy amplification by subsampling becomes essential for achieving the optimal privacy–utility tradeoff. As a result, the final accuracy contains an additional statistical term of order $n^{-1/3}$.

**Estimator decomposition.** For the population analysis, it

is convenient to decompose the error into three parts:

$$
\begin{aligned}
\alpha_t &:= v_t - \nabla_x f(x_t, y_{t+1}), \\
\theta_t &:= u_t - \nabla_y f(x_t, y_t), \qquad\qquad (6) \\
\gamma_t &:= y_t - y^\star(x_t).
\end{aligned}
$$

The first two quantities measure gradient-estimation error (including privacy noise and sampling noise), while $\gamma_t$ measures how well the inner loop tracks the maximizer. These terms enter the value-gradient deviation via

$$
\begin{aligned}
\|v_t - \nabla\Phi(x_t)\| \;\leq\; &\underbrace{\|v_t - \nabla_x f(x_t, y_{t+1})\|}_{\|\alpha_t\|} \\
&+ \underbrace{\|\nabla_x f(x_t, y_{t+1}) - \nabla_x f(x_t, y^\star(x_t))\|}_{\text{controlled by } \|\gamma_t\|}.
\end{aligned}
$$
(7)

**Lemma 2.** *Assume Assumptions 1–4 and 5 hold. Fix $\delta_1 \in (0,1)$ and choose the period $q$, the number of inner steps $K$, and the batch sizes $(S_1, S_2)$ and other parameters as in Appendix B.2. Then, with probability at least $1 - \delta_1$, the iterates produced by Algorithm 2 satisfy, for all outer iterations $t$,*

$$
\|v_t - \nabla\Phi(x_t)\| \;\leq\; \widetilde{\mathcal{O}}\left( \frac{1}{n^{1/3}} + \left( \frac{\sqrt{d\log(1/\delta)}}{n\varepsilon} \right)^{1/2} \right).
$$

Lemma 2 provides a uniform bound on the deviation between the privatized direction $v_t$ and the true value-gradient. This bound is then used in exactly the same two-phase outer-loop analysis as in the ERM case: when $\|v_t\| \geq \alpha$ we obtain a per-step decrease in $\Phi$, while when $\|v_t\| < \alpha$ we invoke the perturb-then-descend escape argument. Failure to escape implies near-PSD curvature at the anchor point.

**Theorem 2.** *Assume Assumptions 1–4 and 5 hold. Fix $(\varepsilon, \delta)$ and failure probabilities $\delta_1, \delta_2 \in (0,1)$. Run Algorithm 1–2 with subsampling-based gradient estimates and Gaussian perturbations calibrated so that the full procedure is $(\epsilon, \delta)$-DP. Choose $(q, S_1, S_2, K)$ as in Appendix B.2 and set the outer-loop parameters $(\eta, \eta_H, r, t_{\mathrm{thres}}, \bar{D})$ according to the same perturbed-descent scaling as in Theorem 1. Then Algorithm 2 outputs a point $x_{\mathrm{out}}$ such that, with probability at least $1 - \delta_1 - \delta_2$,*

$$
\|\nabla\Phi(x_{\mathrm{out}})\| \;\leq\; \alpha, \quad \lambda_{\min}(\nabla^2\Phi(x_{\mathrm{out}})) \;\geq\; -\sqrt{\rho_\Phi\, \alpha},
$$

*with $\alpha = \widetilde{\mathcal{O}}\left( \frac{1}{n^{1/3}} + \left( \frac{\sqrt{d\log(1/\delta)}}{n\varepsilon} \right)^{1/2} \right)$.*

The population guarantee separates two unavoidable sources of error: the statistical term $n^{-1/3}$ arising from stochastic gradients, and the privacy term $(\sqrt{d\log(1/\delta)}/(n\varepsilon))^{1/2}$ arising from protecting the dataset. The key technical distinction from ERM is that Lemma 2 is proved by a period-wise

control of the recursive estimator with subsampling amplification, rather than by summing deviations over the entire optimization horizon.

*Remark* 2. In the non-convex *minimization* literature, (Tao et al., 2025) analyzes first-order private methods and establishes that an $\alpha$-SOSP can be found with $\alpha = \tilde{O}(\frac{1}{n^{1/3}} + (\frac{\sqrt{d}}{n\varepsilon})^{2/5})$. More recently, (Liu & Talwar, 2024) combines the tree mechanism with second-order information to enable saddle-point escape, achieving the improved guarantee $\alpha = \tilde{O}(\frac{1}{n^{1/3}} + (\frac{\sqrt{d}}{n\varepsilon})^{1/2})$. Theorem 2 shows that our minimax framework attains the same population rate $\tilde{O}(\frac{1}{n^{1/3}} + (\frac{\sqrt{d}}{n\varepsilon})^{1/2})$, and hence matches the current ERM state-of-the-art scaling while targeting SOSP of the minimax value function $\Phi$.

*Remark* 3. We have shown that our results match the best-known results in DP-SOSP for ERM in both finite-sum (Liu et al., 2023) and stochastic (Liu & Talwar, 2024) settings. However, the main weakness in these methods is that they need second order (Hessian) information, while our method is purely first-order. To avoid the usage of second order information, we add an inner loop for the update of $y$, which can let us focus on $q$ period instead of the whole iteration $T$. In addition, we set an average moving distance as a critieron for escaping procedure, which facilitates us to update the outer variable $x$ conveniently. Since minimax is more general, as a byproduct, we provide the first first-order DP-SOSP for ERM that matches the best-known results.

## 5. Experiements

### 5.1. Experimental setup

**Problem.** We evaluate our differentially private second-order method on a synthetic low-rank *matrix sensing* instance formulated as a nonconvex–strongly-concave minimax problem. Given measurements $\{(A_i, b_i)\}_{i=1}^n$ with $A_i \in \mathbb{R}^{p \times q}$ and $b_i \in \mathbb{R}$, we optimize a rank-$r$ factorization $X = UV^\top$ ($U \in \mathbb{R}^{p \times r}$, $V \in \mathbb{R}^{q \times r}$) via

$$
\min_{U,V} \max_{y \in \mathbb{R}^n} f(U, V, y) \;:=\; \frac{1}{n} \sum_{i=1}^n \left( y_i(\langle A_i, UV^\top \rangle - b_i) - \tfrac{1}{2} y_i^2 \right).
$$

**Dataset generation.** We set $p = q = 20$, rank $r = 3$, and sample size $n = 400$. Each $A_i \in \mathbb{R}^{p \times q}$ is drawn i.i.d. with entries $A_i(j,k) \sim \mathcal{N}(0, 1/(pq))$. We generate $X^\star = U^\star V^{\star\top}$ with Gaussian factors (followed by a rescaling), and define measurements $b_i = \langle A_i, X^\star \rangle + \xi_i$ with $\xi_i \sim \mathcal{N}(0, \sigma^2)$ and $\sigma = 0.01$. We initialize $U_0, V_0$ with i.i.d. $\mathcal{N}(0, 0.1^2)$ entries and set $y_0 = \mathbf{0}$. Unless otherwise stated, we report a single representative run with seed 0.

**Algorithms and oracle model.** We compare the following DP methods under the same global privacy budget with

| Method | $\Phi(x_{399})$ | $\|\nabla\Phi(x_{399})\|$ | $\lambda_{\min}(\nabla^2\Phi(x_{399}))$ |
|---|---|---|---|
| DP-RGDA (ours) | 0.9119 | 0.4251 | $-4.8504 \times 10^{-2}$ |
| Sto-SPIDER (Liu et al., 2023) | 13.7046 | 3.4505 | $-2.1862 \times 10^{-1}$ |
| Ada-DP-SPIDER (Tao et al., 2025) | 8.0751 | 2.2753 | $-1.4378 \times 10^{-1}$ |
| PrivateDiff (Zhang et al., 2025b) | 0.6546 | 0.3344 | $-4.3622 \times 10^{-2}$ |

*Table 2.* Final objective and stationarity diagnostics on the synthetic matrix sensing instance ($T = 400$). Lower $\Phi$ and $\|\nabla\Phi\|$ are better; $\lambda_{\min}$ closer to 0 indicates milder negative curvature.

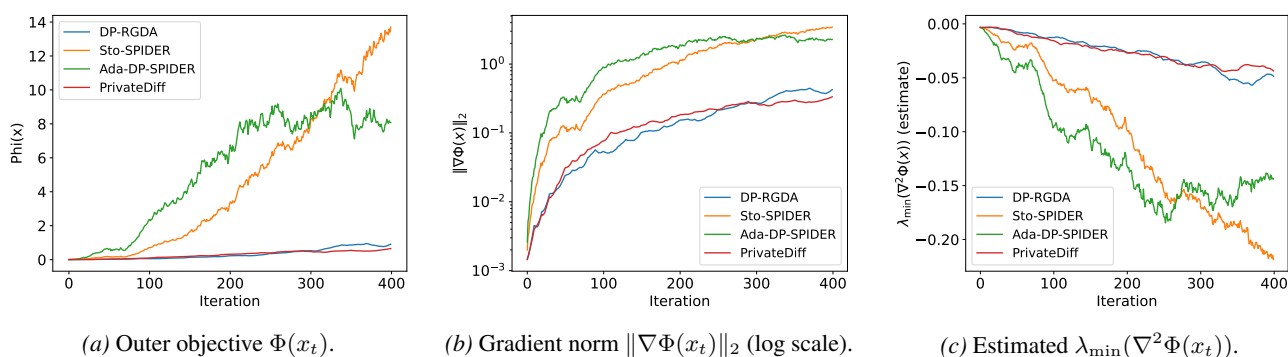

*(a)* Outer objective $\Phi(x_t)$.    *(b)* Gradient norm $\|\nabla\Phi(x_t)\|_2$ (log scale).    *(c)* Estimated $\lambda_{\min}(\nabla^2\Phi(x_t))$.

*Figure 1.* Trajectories on the synthetic matrix sensing minimax instance. DP RGDA implements our SPIDER recursion and (enabled) escape mechanism, while DP-SGDA is a single-loop baseline. We report $\Phi(x_t)$, the (non-private) gradient norm of the induced objective, and an estimated minimum Hessian eigenvalue.

our DP RGDA method. **Sto-SPIDER** (Liu et al., 2023) and **Ada-DP-SPIDER** (Tao et al., 2025) are designed for single-level minimization i.e. DP-SOSP for minimizing an objective. We include them by running them directly on the explicit value function $\Phi(U, V)$ available in this synthetic task, which is a favorable special case of minimax (the inner maximization can be eliminated in closed form). This gives these baselines strictly more permissive access than the general minimax setting, since they do not need to track $y^\star(x)$. **PrivateDiff** (Zhang et al., 2025b), a first-order DP method adapted to the minimax formulation. Unlike DP-RGDA, it targets first-order stationarity and does not include an explicit mechanism aimed at excluding strict saddles of $\Phi$.

**Metrics.** We report (i) the outer objective value $\Phi(x_t)$, (ii) the induced gradient norm $\|\nabla\Phi(x_t)\|$, and (iii) an estimate of the minimum Hessian eigenvalue $\lambda_{\min}(\nabla^2\Phi(x_t))$. The gradient and Hessian-eigenvalue diagnostics are *evaluation only* (computed without privacy noise using the closed-form $\Phi$), and are not used by any method to update iterates.

Due to page limitations, we defer the parameter and privacy settings to Appendix D.

### 5.2. Experimental Results

Figure 1 reports the trajectories of $\Phi(x_t)$, $\|\nabla\Phi(x_t)\|$, and the estimated $\lambda_{\min}(\nabla^2\Phi(x_t))$. Table 2 summarizes the final diagnostics at $t = 399$.

**Comparison to minimization DP-SOSP baselines.** Although Sto-SPIDER and Ada-DP-SPIDER are allowed to optimize the explicit value function $\Phi$ directly (a strictly easier oracle model than general minimax optimization), both baselines are substantially worse than DP-RGDA under the same global privacy budget. At $t = 399$, DP-RGDA achieves $\Phi(x_{399}) \approx 0.912$ and $\|\nabla\Phi(x_{399})\| \approx 0.425$, whereas Sto-SPIDER and Ada-DP-SPIDER remain far from stationarity with $\Phi(x_{399}) \in \{13.70, 8.08\}$ and gradient norms above 2. Moreover, their curvature estimates are more negative ($\lambda_{\min} \approx -0.22$ and $-0.14$), consistent with unstable progress under the compounded DP noise in gradient-difference updates. These results indicate that, in the minimax setting with a fixed privacy budget, our coupled gradient-tracking strategy provides markedly better robustness than directly applying DP-SPIDER-style minimization methods, even on this favorable special case where $\Phi$ is explicit.

**Comparison to a first-order minimax DP baseline.** PrivateDiff achieves a slightly smaller final objective and gradient norm in this particular run. However, this baseline is aligned with first-order optimality and does not incorporate a mechanism explicitly designed to exclude strict saddles of $\Phi$ in general minimax problems. By contrast, DP-RGDA is constructed to target second-order stationarity of the value function via its perturb-and-monitor escape rule while operating under the stricter minimax oracle model (privatized gradients of $f$ and tracked inner maximizers).

Empirically, DP-RGDA attains comparable curvature diagnostics to PrivateDiff (both have mildly negative $\lambda_{\min}$ close to 0), while significantly outperforming the minimization-only DP-SOSP baselines that have more permissive access.

## 6. Conclusion

We develop a first-order method for finding differentially private approximate SOSP for nonconvex–strongly-concave minimax problems. It combines value-function reduction, SPIDER-style gradient tracking, and a perturb-and-monitor escape rule, avoiding Hessians and extra private selection. We prove $(\varepsilon, \delta)$-DP guarantees and utility bounds for both empirical and population objectives, which match the best-known results for DP ERM.

## 7. Acknowledge

Di Wang and Difei Xu are supported in part by the funding BAS/1/1689-01-01,RGC/3/7125-01-01, FCC/1/5940-20-05, FCC/1/5940-06-02, and King Abdullah University of Science and Technology (KAUST) – Center of Excellence for Generative AI, under award number 5940 and a gift from Google.

## Impact Statement

This paper presents algorithmic and theoretical results for computing approximate second-order stationary points in stochastic minimax optimization under differential privacy constraints. The primary positive impact is to enable training and analysis of minimax-based learning procedures on sensitive datasets while providing a formal, quantifiable privacy guarantee for individual data records. Such guarantees can reduce risks of memorization and unintended information leakage compared with non-private optimization, thereby supporting the responsible use of data in high-stakes domains where privacy is essential.

At the same time, differential privacy is not a complete solution to broader issues such as fairness, downstream misuse, or harmful model behaviors, and it can be misunderstood if the privacy parameters $(\varepsilon, \delta)$ are not chosen and communicated carefully. Moreover, privacy-induced noise may degrade utility, which can disproportionately affect settings with limited data and may require careful validation before deployment. Overall, we view the societal consequences of this work as those typical of advances in privacy-preserving machine learning methodology, with the main ethical consideration being correct privacy accounting and clear reporting of the resulting privacy–utility trade-offs.

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

## A. Related Work

**Finding SOSP for Nonconvex Minimax Optimization** The optimization toolbox for finding SOSP is well developed, particularly for methods that leverage second-order information. In recent years, a wide range of algorithms has been introduced for non-convex minimax optimization. Relative to first-order approaches, however, significantly less attention has been devoted to second-order methods for minimax optimization problems with global convergence rate estimation. Meanwhile, a growing body of recent work indicates that first-order stationary points do not ensure local optimality in nonconvex-(strongly) concave settings, nor do they guarantee global optimality in convex-concave settings. In the non-private case, (Lin et al., 2026) proposed newton-based methods that utilize Hessian-vector information, achieving performance that matches the theoretically established lower bound in convex-concave settings. For nonconvex-strongly-concave settings, (Luo et al., 2022) developed Minimax Cubic-Newton, which attains a second-order stationary point of $\Phi$ using Hessian Oracles. Additionally, (Yang et al., 2023) proposed a Perturbed Restarted Accelerated HyperGradient Descent algorithm, improving the complexity bound using only gradient iterations. However, due to the large noise needed to add to the Hessian, it will be sub-optimial to direclt privatize these methods.

**DP Minimax** As privacy concerns around data have grown, differential privacy (DP) has become an essential requirement in stochastic optimization. The study of first-order stationary points traces back to the seminal work of (Dwork et al., 2006), which established the foundational framework of differential privacy. Subsequent research has significantly expanded its role in stochastic optimization. DP minimax optimization has also developed rapidly (Yang et al., 2022; Zhang et al., 2025b; 2022; Zhao et al., 2023). For example, (Yang et al., 2022) investigates DP-SGDA for stochastic minimax learning via an algorithmic-stability perspective and derives a near-optimal guaranty for convex-concave objectives using SGDA. Moving beyond convex/PL regimes, (Zhang et al., 2025b) studies differentially private stochastic minimax optimization in the nonconvex-strongly-concave (NC-SC) setting and provides the first general results in this direction.

**DP SOSP** However, research on first-order stationary points (FOSP) is insufficient because FOSPs can be local minima, saddle points, or even local maxima. Therefore, finding second-order stationary points (SOSP) has become a crucial problem in nonconvex optimization. Related progress has also been made for private non-convex minimization when the goal is to reach SOSP. In particular, (Tao et al., 2025) considers gradient-based procedures for finding the SOSP for a minimum problem and claims that an $\alpha$-SOSP can be achieved with $\alpha = \tilde{O}(\frac{1}{n^{1/3}} + (\frac{\sqrt{d}}{n\varepsilon})^{2/5})$. Furthermore, (Liu & Talwar, 2024) connects the guarantees for finding first-order stationary points and second-order stationary points, obtaining $\alpha = \tilde{O}(\frac{1}{n^{1/3}} + (\frac{\sqrt{d}}{n\varepsilon})^{1/2})$ for an $\alpha$-SOSP. Their approach combines the Tree mechanism with second-order information to facilitate escape from saddle points. By contrast, our methods use only first-order information, yet attain the same $\alpha$-SOSP rate as in (Liu & Talwar, 2024).

## B. Proof

### B.1. Proof of the Empirical Loss

To simplify the discussion below, we define $\nabla_t = (\frac{1}{S_1}\sum_{i=1}^{S_1}\nabla_x F(x_t, y_t; \xi_i) + \omega_t, -[\frac{1}{S_1}\sum_{i=1}^{S_1}\nabla_y F(x_t, y_t; \xi_i) + \tau_t])$ and $\Delta_t = (v_{t,k}, -u_{t,k})$. Note that we need to set $K = 1$ for ERM so there will not be any confusion for the index $k$, namely, the subscript of $\Delta_t$ is well-defined. By Proposition 1 in (Fang et al., 2018), we have the following lemma:

**Lemma 3.** *Consider Algorithm 3, and for any $t \in \{0, ..., T\}$ let $t_0 = \left\lfloor \frac{t}{q} \right\rfloor q$. If each $\nabla_t$ computed defined above is an unbiased estimate of $\nabla F(w_t; S)$ satisfying with probability $1 - \delta_1$,*

$$\|\nabla_{t_0} - \nabla F(w_{t_0}; S)\|^2 \leq B_1^2 \log(1/\delta_1),$$

*and each $\Delta_t$ is an unbiased estimate of the gradient variation satisfying*

$$\|\Delta_t - [\nabla F(w_t; S) - \nabla F(w_{t-1}; S)]\|^2 \leq B_2^2 \|w_t - w_{t-1}\|^2 \log(1/\delta_1).$$

*Then for any $t \geqslant t_0 + 1$, the iterates of Algorithm satisfy*

$$\|\nabla_t - \nabla F(w_t)\|^2 \leq \log(1/\delta)(B_2^2 \sum_{k=t_0+1}^{t} \|w_k - w_{k-1}\|^2 + B_1^2).$$

**Lemma 4.** *Let the conditions of Lemma 1 be satisfied. Let $\eta \leq \frac{1}{2L_1}$ and $q \leq O\left(\frac{1}{T^2\eta^2}\right)$. Then the output of Inner Updater, $\bar{w}$, satisfies*

$$\mathbb{E}[\|\nabla F(\bar{w}; S)\|] = O\left(\sqrt{\frac{F_0}{qT} + B_1}\right). \tag{8}$$

*Proof.* In the following, for any $t \in [T]$, let $t_0 = \left\lfloor \frac{t}{q} \right\rfloor q$ (i.e. the index corresponding to the start of the phase containing iteration $t$). To simplify the notation, we define $w_t = (x_t, y_t)$.

By a standard analysis for smooth functions we have (recalling that $\nabla_t$ is an unbiased estimate of $\nabla F(w_t; S)$ for any $t \in [T]$)

$$F(w_{t+1}; S) \leq F(w_t; S) + \frac{\eta}{2}\|\nabla F(w_t; S) - \nabla_t\|^2 - \left(\frac{\eta}{2} - \frac{L_1\eta^2}{2}\right)\|\nabla_t\|^2. \tag{9}$$

Taking expectation we have the following manipulation using the update rule of Algorithm 2

$$F(w_{t+1}; S) - F(w_t; S)$$
$$\leq \log(1/\delta_1)(\frac{\eta}{2}\|\nabla F(w_t; S) - \nabla_t\|^2 - \left(\frac{\eta}{2} - \frac{L_1\eta^2}{2}\right)\|\nabla_t\|^2)$$
$$\leq \log(1/\delta_1)(\frac{\eta B_2^2}{2} \sum_{k=t_0+1}^{t} \|w_{k+1} - w_k\|^2 + \frac{\eta}{2}\|\nabla_{t_0} - F(w_{t_0}; S)\|^2 - \left(\frac{\eta}{2} - \frac{L_1\eta^2}{2}\right)\|\nabla_t\|^2)$$
$$\leq \log(1/\delta_1)(\frac{\eta^3 B_2^2}{2} \sum_{k=t_0+1}^{t} \|\nabla_k\|^2 + \frac{\eta B_1^2}{2} - \left(\frac{\eta}{2} - \frac{L_1\eta^2}{2}\right)\|\nabla_t\|^2),$$

where the second inequality follows from Lemma 4.1 and the last inequality follows from the update rule. Note that if $t = t_0$ the sum is empty. Summing over a given phase we have

$$F(w_{t+1}; S) - F(w_{t_0}; S) \leq \log(1/\delta_1)(\frac{\eta^3 B_2^2}{2} \sum_{k=t_0}^{t} \sum_{j=t_0+1}^{k} \|\nabla_j\|^2 + \sum_{k=t_0}^{t} \left[\frac{\eta B_1^2}{2} - \left(\frac{\eta}{2} - \frac{L_1\eta^2}{2}\right)\|\nabla_k\|^2\right])$$
$$\leq \log(1/\delta_1)(\frac{\eta^3 B_2^2 q}{2} \sum_{k=t_0}^{t} \|\nabla_k\|^2 + \sum_{k=t_0}^{t} \left[\frac{\eta B_1^2}{2} - \left(\frac{\eta}{2} - \frac{L_1\eta^2}{2}\right)\|\nabla_k\|^2\right])$$
$$= \log(1/\delta_1)(-\sum_{k=t_0}^{t} \underbrace{\left(\frac{\eta}{2} - \frac{L_1\eta^2}{2} - \frac{\eta^3 B_2^2 q}{2}\right)}_{A_0}\|\nabla_k\|^2 - \frac{\eta B_1^2}{2}),$$

where the second inequality comes from the fact that each gradient appears at most $q$ times in the sum. We now sum over all phases. Let $P = \{p_0, p_1, ..., \} = \left\{0, q, 2q, ..., \left\lfloor \frac{T-1}{q} \right\rfloor q, T\right\}$. We have

$$F(w_T; S) - F(w_0; S) \leq \sum_{i=1}^{|P|}(F(w_{p_i}; S) - F(w_{p_{i-1}}; S))$$
$$\leq \log(1/\delta_1)(-\sum_{t=0}^{T} A_0 \mathbb{E}\left[\|\nabla_t\|^2\right] + \frac{T\eta B_1^2}{2}).$$

Rearranging the above yields

$$\frac{1}{T}\sum_{t=0}^{T}\|\nabla_k\|^2 \le \frac{1}{\log(1/\delta_1)}\frac{F_0}{TA_0} + \frac{\eta B_1^2}{2A_0}. \tag{10}$$

Now let $i^*$ denote the index of $\bar{w}$ selected by the algorithm. Note that

$$\|\nabla F(w_{i^*};S)\|^2 \le 2\|\nabla F(w_{i^*};S) - \nabla_{i^*}\|^2 + 2\|\nabla_{i^*}\|^2. \tag{11}$$

The second term above can be bounded via inequality 10. To bound the first term we have by Lemma 1 that

$$\|\nabla_{i^*} - \nabla F(w_{i^*};S)\|^2 \le \log(1/\delta_1)(B_2^2\sum_{k=s_{i^*}+1}^{i^*}\|w_k - w_{k-1}\|^2 + B_1^2)$$

$$= \log(1/\delta_1)(\eta^2 B_2^2\sum_{k=s_{i^*}+1}^{i^*}\|\nabla_k\|^2 + B_1^2)$$

$$\le \log(1/\delta_1)(\frac{q\eta^2 B_2^2}{T}\sum_{k=0}^{T}\|\nabla_k\|^2 + B_1^2)$$

$$\le \log(1/\delta_1)(\frac{B_2^2\eta^2 qF_0}{TA_0} + \frac{\eta^3 qB_2^2}{2A_0}B_1^2 + B_1^2),$$

where the last inequality comes from inequality (3) and the expectation over $i^*$. Plugging into inequality (4) one can obtain

$$\|\nabla F(w_{i^*};S)\|^2 \le \log(1/\delta_1)(\frac{2F_0}{TA_0}(1 + B_2^2\eta^2 q) + \left(\frac{\eta}{A_0} + 2 + \frac{B_2^2\eta^3 q}{A_0}\right)B_1^2). \tag{12}$$

Now recall $A_0 = \frac{\eta}{2} - \frac{L_1\eta^2}{2} - \frac{\eta^3 B_2^2 q}{2}$. Since $q \le O\left(\frac{1}{B_2^2\eta^2}\right)$ and $\eta \le \frac{1}{2L_1}$ we have $A_0 = \Theta(\eta)$. Thus plugging into inequality (5) and again using the fact that $q \le O\left(\frac{1}{B_2^2\eta^2}\right)$ we have

$$\|\nabla F(w_{i^*};S)\|^2 = O\left(\frac{F_0}{T\eta}(1 + B_2^2\eta^2 q) + \left(3 + \frac{B_2^2\eta^3 q}{A_0}\right)B_1^2\right) = O\left(\frac{F_0}{T\eta} + B_1^2\right).$$

The claim then follows from the Jensen inequality. $\qquad\square$

Given the above lemma, we can derive the following theorem:

**Lemma 5.** *For* $\eta = \frac{1}{2L}$, *we set* $S_1 = n$ *and* $S_2 \ge \max\{(\frac{Mn\varepsilon}{\sqrt{F_0 Ld\log(1/\delta)}})^{2/3}, \frac{(Mnd\log(1/\delta))^{1/3}}{(LF_0)^{1/6}}\}$, $T = \max\{(\frac{(LF_0)^{1/4}}{\sqrt{M\bar{\epsilon}}})^{4/3}, \frac{n\varepsilon}{\sqrt{d\log(1/\delta)}}\}$ *and* $q = \lfloor\frac{n^2\varepsilon^2}{L^2 Td\log(1/\delta)}\rfloor$. *With probability at least* $1 - \delta_1$ *we have*

$$\|\nabla F(x,y)\| \le \tilde{\mathcal{O}}((\frac{\sqrt{d\log(1/\delta)}}{n\varepsilon})^{2/3}).$$

*Proof.* **Privacy Proof**: Note that each gradient estimate computed in Algorithm 2 is $M$ and this estimate is computed at most $\frac{T}{q}$ times. Similarly, for the gradient variation at step $t$, we have norm bound $L\|w_t - w_{t-1}\| = L\|y_t - y_{t-1}\|$ and computed at most $T$ times. Therefore, the scale of noise in both cases ensures the overall algorithm is $(\varepsilon, \delta)$-DP.

**Convergence proof**: By Lemma 4, after setting the parameters, all we need to determine is to specify $B_1$ and $B_2$. To simplify the notation in the following discussion, we define $\bar{\epsilon} = \frac{\sqrt{d\log(1/\delta)}}{n\varepsilon}$. We pick $S_1 = n$, then the condition of Lemma 1 are satisfied with $B_1^2 = \mathcal{O}(\frac{T\bar{\epsilon}^2}{q})$ and $B_2^2 = \mathcal{O}(\frac{L^2}{S_2} + L^2 T\bar{\epsilon}^2)$. Our parameter setting needs to guarantee that $T \ge q$ and $T \ge \frac{n^2}{S_2^2}$. To make $B_2$'s expression consistent, we set $S_2 \ge \frac{1}{T\bar{\epsilon}^2}$. Therefore, $B_2 = \mathcal{O}(T\bar{\epsilon}^2)$. Thus, the condition on Lemma

4 is satisfied with $q = \frac{L_1^2}{B_2^2} = \frac{1}{T\bar{\epsilon}^2}$ by setting $\eta = \frac{1}{2L}$. Then combine the above discussion with Lemma 4 with probability at least $1 - \delta_1$, we have

$$\|\nabla F(x, y)\| = \mathcal{O}(\sqrt{\frac{F_0 L}{T}} + \frac{M\sqrt{T}\bar{\epsilon}}{\sqrt{q}})$$

$$= \mathcal{O}(\sqrt{\frac{F_0 L}{T}} + MT\bar{\epsilon}^2)$$

Now let's turn to $T$. Due to the setting $q = \frac{1}{T\bar{\epsilon}^2}$, it suffices to set $T \geqslant \frac{1}{\bar{\epsilon}}$. Therefore, we can set $T = \max\{(\frac{(LF_0)^{1/4}}{\sqrt{M}\bar{\epsilon}})^{4/3}, \frac{1}{\bar{\epsilon}}\}$. After setting the above parameters, we can derive that

$$\|\nabla F(x, y)\| \leqslant \mathcal{O}((\sqrt{F_0}\bar{\epsilon})^{2/3}) = \mathcal{O}((\frac{\sqrt{d \log(1/\delta)}}{n\varepsilon})^{2/3}).$$

To derive the above rate, there still are some parameters to be clarified. The restrictions on the batch size implied by $T$ indicate that $S_2 \geqslant \frac{n}{\sqrt{T}}$ and thus to have $S_2 \geqslant \frac{M^{1/3} n\bar{\epsilon}^{2/3}}{(LF_0)^{1/6}}$ to satisfy the setting of $T$. Recall that we also need that $S_2 \geqslant \frac{1}{T\bar{\epsilon}^2}$, thus we need $S_2 \geqslant (\frac{M}{\sqrt{F_0 L}\bar{\epsilon}})^{2/3}$.

$\square$

**Lemma 6.** *Given the parameter setting as follows, stepsize $\eta_H$, escaping threshold $t_{thres} = 2\log(\frac{\eta_H \alpha_H L_\Phi}{C\rho_\Phi r_0})/\eta_H = \tilde{O}(\frac{1}{\eta_H \alpha_H})$, perturbation radius $r \leqslant R$ and average movement $\bar{D} \leqslant R^2/t_{thres}^2$ and $\alpha_H = \sqrt{\rho_\Phi \alpha}$. Then for $\forall s$, if our algorithm does not break the escaping phase, then we have $\lambda_{\min} \geqslant -\alpha_H$ with probability $1 - \delta_1 - \delta_2$.*

*Proof.* Let $\{x_t\}, \{x_t'\}$ be two coupled sequences by running 2 algorithm from $x_{m_s+1} = x_{m_s} + \xi$ and $x'_{m_s+1} = x'_{m_s} + \xi'$ with $x_{m_s+1} - x'_{m_s+1} = r_0 e_1$, where $\xi, \xi' \in B_0(r)$, $r_0 = \frac{\delta_2 r}{\sqrt{d}}$ and $e_1$ denotes the smallest eigenvector direction of $\nabla^2\Phi(x_{m_s})$. When $\lambda_{\min}(\nabla^2\Phi(x_{m_s})) \leqslant -(\frac{\sqrt{d \log(1/\delta)}}{n\varepsilon})$, by Lemma 7 we have

$$\max_{m_s < t \leqslant m_s + t_{thres}} \{\|x_t - x_{m_s}\|, \|x_t' - x_{m_s}\|\} \geqslant R$$

with probability at least $1 - 4\delta_1$.

Let $\mathcal{S}$ be the set of $x_{m_s+1}$ that will not generate a sequence moving out of the ball with center $x_{m_s}$ and radius $r$. Then the projection of $\mathcal{S}$ onto direction $e_1$ should not be larger than $r_0$. By integration, we can calculate the volume of the ball and stuck region in $d$-dimension and further check that the probability of $x_{m_s+1} \in \mathcal{S}$ is smaller than $\delta_2$ as $\xi$ is drawn from uniform distribution, which is shown in Eq. (13):

$$Pr(x_{m_s+1} \in \mathcal{S}) \leqslant \frac{r_0 V_{d-1}(r)}{V_d(r)} \leqslant \frac{\sqrt{d} r_0}{r} \leqslant \delta_2 \tag{13}$$

where $V_d(r)$ is the volume of $d$-dimension ball with radius $r$. Applying union bound, with probability at least $1 - 4\delta_1 - \delta_2$ we have

$$\exists m_s < t \leqslant m_s + t_{thres}, \quad \|x_t - x_{m_s+1}\| \geqslant R. \tag{14}$$

If Algorithm 2 does not break the escaping phase, then for $\forall m_s < t \leqslant m_s + t_{thres}$ we have

$$\|x_t - x_{m_s+1}\| < \sqrt{(t - m_s) \sum_{i=m_s+1}^{t-1} \|x_{i+1} - x_i\|^2} \leqslant (t - m_s)\sqrt{\bar{D}} \tag{15}$$

which is derived by Cauchy-Schwartz inequality. By the choice of parameters $t_{thres}$ and $\bar{D}$, we have

$$\|x_t - x_{m_s+1}\| < t_{thres}\sqrt{\bar{D}} \leqslant R. \tag{16}$$

Therefore, when $\lambda_{\min}(\nabla^2\Phi(x_{m_s})) \leqslant -\alpha_H$, with probability at least $1 - 4\delta_1 - \delta_2$ our Algorithm 2 will break the escaping phase.

$\square$

**Lemma 7.** *Set stepsize $\eta_H \leqslant \frac{r_0}{2\|v_t\|} \leqslant \frac{1}{2}(\frac{\sqrt{d\log(1/\delta)}}{n\varepsilon})^{-4/3}r_0$, and $R = \frac{1}{2L_\Phi \eta_H} = \frac{1}{L_\Phi r_0}(\frac{\sqrt{d\log(1/\delta)}}{n\varepsilon})^{4/3}$, perturbation radius $r \leqslant \frac{L_\Phi \eta_H \alpha_H}{C\rho_\Phi}$ and threshold $t_{thres} = 2\log(\frac{\eta_H \alpha_H L_\Phi}{C\rho_\Phi r_0})/\eta_H = \tilde{O}(\frac{1}{\eta_H \alpha_H})$, where $r_0 \leqslant r$ and $C = \tilde{O}(1)$. Suppose $-\lambda_{\min}(\nabla^2\Phi(x_{m_s})) \leqslant -\alpha_H$.*

*Let $\{x_t\}, \{x'_t\}$ be two coupled sequences by running Algorithm 2 from $x_{m_s+1} = x_{m_s} + \xi$ and $x'_{m_s+1} = x_{m_s} + \xi'$ with $x_{m_s+1} - x'_{m_s+1} = r_0 e_1$, where $\xi, \xi' \in B_0(r)$ and $e_1$ denotes the smallest eigenvector direction of $\nabla^2\Phi(x_{m_s})$. Then with probability at least $1 - 4\delta_1$ (for $\delta_1$ in Lemma 4), we have*

$$\max_{m_s < t \leqslant m_s + t_{thres}} \{\|x_t - x_{m_s}\|, \|x'_t - x_{m_s}\|\} \geqslant R. \tag{17}$$

*Proof.* To prove this lemma, we assume the contrary:

$$\forall m_s < t \leqslant m_s + t_{\text{thres}}, \quad \|x_t - x_{m_s}\| < R, \quad \|x'_t - x_{m_s}\| < R. \tag{18}$$

Define $w_t = x_t - x'_t$ and $\nu_t = v_t - \nabla\Phi(x_t) - (v'_t - \nabla\Phi(x'_t))$. We have

$$\begin{aligned} w_{t+1} &= w_t - \eta_H(v_t - v'_t) = w_t - \eta_H(\nabla\Phi(x_t) - \nabla\Phi(x'_t)) - \eta_H \nu_t \\ &= (I - \eta_H \mathcal{H})w_t - \eta_H(\Delta_t w_t + \nu_t) \end{aligned} \tag{19}$$

where

$$\mathcal{H} = \nabla^2\Phi(x_{m_s}), \qquad \Delta_t = \int_0^1 \left[\nabla^2\Phi(x'_t + \theta(x_t - x'_t)) - \mathcal{H}\right]d\theta. \tag{20}$$

Let

$$p_{t+1} = (I - \eta_H \mathcal{H})^{t-m_s} w_{m_s+1}, \qquad q_{t+1} = \eta_H \sum_{\tau=m_s+1}^{t} (I - \eta_H \mathcal{H})^{t-\tau}(\Delta_\tau w_\tau + \nu_\tau) \tag{21}$$

and apply recursion to Eq. (19), we can obtain

$$w_{t+1} = p_{t+1} - q_{t+1}. \tag{22}$$

Next, we will inductively prove

$$\|q_t\| \leqslant \|p_t\|/2, \quad \forall m_s < t \leqslant m_s + t_{\text{thres}}. \tag{23}$$

First, when $t = m_s + 1$ the conclusion holds since $\|q_{m_s+1}\| = 0$. Suppose the above equation is satisfied for $\tau \leqslant t$. Then we have

$$\|w_\tau\| \leqslant \|p_\tau\| + \|q_\tau\| \leqslant \tfrac{3}{2}\|p_\tau\| = \tfrac{3}{2}(1 + \eta_H \gamma)^{\tau-m_s-1}r_0. \tag{24}$$

Then for the case $\tau = t + 1$, by the above two equations we have

$$\begin{aligned} \|q_{t+1}\| &\leqslant \eta_H(1 + \eta_H \gamma)^{t-m_s} \cdot \frac{3}{2}\sum_{\tau=m_s+1}^{t} \|\Delta_\tau\| r_0 + \eta_H \sum_{\tau=m_s+1}^{t}(1 + \eta_H \gamma)^{t-\tau}\|\nu_\tau\| \\ &\leqslant (1 + \eta_H \gamma)^{t-m_s}\left(\eta_H L_\Phi R r_0 + \tfrac{1}{4}r_0\right) \\ &\leqslant \tfrac{1}{2}(1 + \eta_H \gamma)^{t-m_s}r_0 = \|p_{t+1}\|/2. \end{aligned} \tag{25} \tag{26}$$

to get the above result, all we need is to assume that $L_\Phi \eta_H R \leqslant \frac{3}{4}$.

In the second inequality, we use Lipschitz Hessian to obtain $\|\Delta_\tau\| \leqslant L_\Phi R$ and we use Lemma 4 problem and the fact

$$a^{t+1} - 1 = (a - 1)\sum_{s=0}^{t} a^s$$

to obtain $\|\nu_\tau\| \leqslant (\frac{\sqrt{d\log(1/\delta)}}{n\varepsilon})^{2/3}$ with probability $1 - 4\delta_1$. The last inequality can be achieved by the definitions of $\eta_H$ and $t_{\text{thres}}$.

Now we have

$$\frac{1}{2}(1 + \eta_H\gamma)^{t-m_s-1}r_0 \leqslant \|w_t\| \leqslant \|x_t - x_{m_s}\| + \|x_t' - x_{m_s}\| \tag{27}$$

which conflicts with Eq. (18) due to the choice of $t_{\text{thres}}$.

Note that $e_1$ is the eigenvector of Hessian H, i.e. $He_1 = -re_1$. Therefore, $P_{t+1} = (I - \eta_H H)^{t-m_s}$ $\qquad\square$

**Theorem 3.** *Combining the above two lemmas, we can finally derive that Algorithm 2 is $(\varepsilon, \delta)$-DP and outputs a point $x_{\text{out}}$ such that, with probability at least $1 - \delta_1 - \delta_2$,*

$$\|\nabla\Phi_S(x_{\text{out}})\| \leqslant \alpha, \lambda_{\min}(\nabla^2\Phi_S(x_{\text{out}})) \geqslant -\sqrt{\rho_\Phi\,\alpha},$$

*with $\alpha = \widetilde{\mathcal{O}}(\bar{\epsilon}^{2/3}) = \widetilde{\mathcal{O}}\left(\left(\frac{\sqrt{d\log(1/\delta)}}{n\varepsilon}\right)^{2/3}\right).$*

## B.2. Proof of the loss in Population

Before we start our proof, first, we define the following notations.

$$\mathcal{G}_\lambda(x,y) = \frac{y - \Pi_{\mathcal{Y}}(y + \lambda\nabla_y f(x,y))}{\lambda}, \quad \gamma_t = \mathcal{G}_\lambda(x_t, y_{t+1}), \tag{28}$$

$$\alpha_t = v_t - \nabla_x f(x_t, y_{t+1}), \quad \theta_t = u_t - \nabla_y f(x_t, y_{t+1}). \tag{29}$$

**Lemma 8.** *Set stepsize* $\eta \leqslant \frac{1}{L\log(4/\delta_1)C_1}(\frac{\sqrt{d\log(1/\delta)}}{n\varepsilon})^{1/2}$, $\lambda = \frac{1}{6L}$, *batchsize* $S_1 = n$ *and* $S_2 \geqslant \log^2(4/\delta_1)\kappa(\frac{n\varepsilon}{\sqrt{d\log(1/\delta)}})^{1/3}$, *period* $q = O((\frac{\sqrt{d\log(1/\delta)}}{n\varepsilon})^{1/3})$, *inner loop* $K = O(\kappa)$, *perturbation radius* $r \leqslant \frac{1}{L\log(4/\delta_1)}(\frac{\sqrt{d\log(1/\delta)}}{n\varepsilon})^{1/2}$ *and average movement* $D \leqslant \frac{1}{L^2\log^2(4/\delta_1)C_1^2}(\frac{\sqrt{d\log(1/\delta)}}{n\varepsilon})$, *where $C_1 = O(1)$ is a constant to be decided later. The initial value of $y_0$ satisfies $\|G_\lambda(x_0, y_0)\| \leqslant 1/\kappa(\frac{\sqrt{d\log(1/\delta)}}{n\varepsilon})^{1/2}$. With probability at least $1 - 4\delta_1$, for $\forall t$ we have $\|\alpha_t\| \leqslant \mathcal{O}(\frac{1}{n^{1/3}} + (\frac{\sqrt{d\log(1/\delta)}}{n\varepsilon})^{1/2})$, $\|\theta_t\| \leqslant \mathcal{O}(\frac{1}{n^{1/3}} + (\frac{\sqrt{d\log(1/\delta)}}{n\varepsilon})^{1/2})$ and $\|\gamma_t\| \leqslant \mathcal{O}(\frac{1}{n^{1/3}} + (\frac{\sqrt{d\log(1/\delta)}}{n\varepsilon})^{1/2})$. Moreover, we have $\|v_t - \nabla\Phi(x_t)\| \leqslant \mathcal{O}(\frac{1}{n^{1/3}} + (\frac{\sqrt{d\log(1/\delta)}}{n\varepsilon})^{1/2})$.*

*Proof.* **Privacy Proof:** For the adjoint datasets $\mathcal{D}$ and $\mathcal{D}$' have $x$ and $x'$ in $q$-th position in difference. By the Lipschitzness and Lipschitz gradient assumption, we can derive that

$$\Delta_1 = \|v_t - v_t'\| = \frac{1}{S_1}\|(\nabla_x F(x,y;\xi) - \nabla_x F(x,y;\xi'))\| \leqslant \frac{2M}{S_1} \tag{30}$$

$$\Delta_2 = \|v_t - v_t'\| \leqslant \frac{1}{S_2}\|(\nabla_x F(x,y_k,\xi) - \nabla_x F(x,y_{k-1},\xi)) - (\nabla_x F(x,y_k;\xi;) - \nabla_x F(x,y_{k-1};\xi'))\| \leqslant \frac{2\sqrt{2}L\|y_k - y_{k-1}\|}{S_2} \tag{31}$$

Similarly, the same sensitivity holds for the partial derivative with respect to $y$.

By the Gaussian mechanism, as long as we add noise with variance $\sigma_{\omega_t} = \sigma_{\tau_t} = \frac{C_1 M\sqrt{\log(1.25/\delta)}}{S_1\varepsilon}$ and $\sigma_{\zeta_t} = \sigma_{\chi_t} = \frac{2C_2 Lq_t\sqrt{\log(1.25/\delta)}}{S_2\varepsilon}\|y_{t,k} - y_{t,k-1}\| = \frac{2C_2 L\sqrt{\log(1.25/\delta)}}{n\varepsilon}\|y_{t,k} - y_{t,k-1}\|$, where we define the sampling rate $q_t = S_2/n$ then we can guarantee that the query for $u_t$ and $v_t$ are $(\varepsilon/2, \delta)$-DP respectively. Therefore, by the composition theorem, Algorithm 2 satisfies $(\varepsilon, \delta)$-DP.

**Convergence Proof:**

First, we define the following notations.

$$G_\lambda(x,y) = \frac{y - \Pi_{\mathcal{Y}}(y + \lambda\nabla_y f(x,y))}{\lambda}, \quad \gamma_t = G_\lambda(x_t, y_{t+1}),$$

$$\alpha_t = v_t - \nabla_x f(x_t, y_{t+1}), \quad \theta_t = u_t - \nabla_y f(x_t, y_{t+1}) \tag{32}$$

Then we have the following estimation of $\alpha_t$, $\theta_t$ and $\gamma_t$ in Lemma 8 to show their magnitude are bounded by $O(\kappa^{-1}\alpha)$ and $\|v_t - \nabla\Phi(x_t)\|$ is bounded by $O(\alpha)$.

According to the definition of $\alpha_t$ and $\theta_t$, when $\mathrm{mod}(t+1, q) \neq 0$ and suppose that $\|\chi_t\|$ can be bounded by $B_{\chi_t}$ with probability $1 - \delta_1$, we have

$$
\alpha_{t+1} - \alpha_t = \frac{1}{S_2} \sum_{k=1}^{s_t} \sum_{i=1}^{S_2} \left( \nabla_x F(x_{t+1}, y_{t+1,k}; \xi_{k,i}) - \nabla_x F(x_{t+1}, y_{t+1,k-1}; \xi_{k,i}) \right)
$$

$$
- \left( \nabla_x f(x_{t+1}, y_{t+1,k}) - \nabla_x f(x_{t+1}, y_{t+1,k-1}) \right) + \frac{1}{S_2} \sum_{i=1}^{S_2} \nabla_x F(x_{t+1}, y_{t+1}; \xi_i)
$$

$$
- \nabla_x F(x_t, y_{t+1}; \xi_i) - \left( \nabla_x f(x_{t+1}, y_{t+1}) - \nabla_x f(x_t, y_{t+1}) \right) + \sum_{k=1}^{s_t} \zeta_{t,k},
$$

$$
\theta_{t+1} - \theta_t = \frac{1}{S_2} \sum_{k=1}^{s_t} \sum_{i=1}^{S_2} \left( \nabla_y F(x_{t+1}, y_{t+1,k}; \xi_{k,i}) - \nabla_y F(x_{t+1}, y_{t+1,k-1}; \xi_{k,i}) \right)
$$

$$
- \left( \nabla_y f(x_{t+1}, y_{t+1,k}) - \nabla_y f(x_{t+1}, y_{t+1,k-1}) \right) + \frac{1}{S_2} \sum_{i=1}^{S_2} \nabla_y F(x_{t+1}, y_{t+1}; \xi_i)
$$

$$
- \nabla_y F(x_t, y_{t+1}; \xi_i) - \left( \nabla_x f(x_{t+1}, y_{t+1}) - \nabla_y f(x_t, y_{t+1}) \right) + \sum_{k=1}^{s_t} \chi_{t,k}.
$$

By the property of sub-Gaussian variable and union bound together with the property of sub-Gaussian, for $\forall t$, with probability at least $1 - 2\delta_1$ we have

$$
\|\alpha_{t+1}\|^2 \leqslant 4\log(4/\delta_1) \left( \frac{\sigma^2}{S_1} + \sigma_{\omega_{\lfloor t/q \rfloor q}}^2 + \frac{4L^2}{S_2} \sum_{i=[t/q]q}^{t} \left( \|x_{i+1} - x_i\|^2 + \sum_{k=1}^{s_i} \|y_{i+1,k} - y_{i+1,k-1}\|^2 \right) \right)
$$

$$
+ \log(1/(2\delta_1)) \left( \sum_{i=\lfloor t/q \rfloor q}^{t} \sum_{k=1}^{s_i} \sigma_{\xi_{i,k}}^2 \right)
$$
(33)

$$
\|\theta_{t+1}\|^2 \leqslant 4\log(4/\delta_1) \left( \frac{\sigma^2}{S_1} + \sigma_{\tau_{\lfloor t/q \rfloor q}}^2 + \frac{4L^2}{S_2} \sum_{i=\lfloor t/q \rfloor q}^{t} \left( \|x_{i+1} - x_i\|^2 + \sum_{k=1}^{s_i} \|y_{i+1,k} - y_{i+1,k-1}\|^2 \right) \right)
$$

$$
+ \log(1/(2\delta_1)) \left( \sum_{i=\lfloor t/q \rfloor q}^{t} \sum_{k=1}^{s_i} \sigma_{\chi_{i,k}}^2 \right)
$$
(34)

Then by Lemma 11 we can obtain

$$
\|y_{t,k+1} - y_{t,k}\|^2 \leqslant \|y_{t,k} - y_{t,k-1}\|^2 - 2\lambda \left[ \frac{\mu L}{\mu + L} \|y_{t,k} - y_{t,k-1}\|^2 + \frac{1}{\mu + L} \left\| \frac{1}{S_2} \sum_{i=1}^{S_2} (\nabla_y F(x_t, y_{t,k}) - \nabla_y F(x_t, y_{t,k-1})) \right\|^2 \right]
$$

$$
+ \frac{\lambda^2 L^2}{S_2} \|y_{t,k} - y_{t,k-1}\|^2 + \lambda^2 \sigma_{\chi_{t,k}}^2 + 2\lambda \langle y_{t,k} - y_{t,k-1}, \chi_{t,k} \rangle.
$$

Then summing over $k$, with probability at least $1 - \delta_1$, we have

$$
\sum_{k=1}^{s_t-1} \|y_{t,k} - y_{t,k-1}\|^2 \leqslant \sum_{k=1}^{s_t-1} \left( 1 - \frac{2\lambda\mu L}{\mu + L} + \frac{\lambda^2 L^2}{S_2} \right) \|y_{t,k} - y_{t,k-1}\|^2 + A_\sigma \sum_{k=1}^{s_t-1} \|y_{t,k} - y_{t,k-1}\| + \lambda^2 \log(1/\delta_1) \sum_{k=1}^{s_t-1} \sigma_{\chi_{t,k}}^2,
$$

where we denote $A_\sigma = 2\sqrt{2}\sigma_{\chi_{t,k}}\sqrt{\log(1/\delta_1)}$.

A little adjustment gives

$$\sum_{k=1}^{s_t-1} \|y_{t,k+1} - y_{t,k}\|^2 \leqslant \sum_{k=0}^{s_t-1} \left(1 - \frac{2\lambda\mu L}{\mu+L} + \frac{\lambda^2 L^2}{S_2}\right) \|y_{t,k+1} - y_{t,k}\|^2 + \lambda^2 \log(1/\delta_1) \sum_{k=1}^{s_t-1} \sigma_{\chi_{t,k}}^2.$$

Implement transformation from both sides, then we can get

$$\sum_{k=1}^{s_t-1} \left(\frac{2\lambda\mu L}{\mu+L} - \frac{\lambda^2 L^2}{S_2}\right) \|y_{t,k+1} - y_{t,k}\|^2 \leqslant \|y_{t,1} - y_{t,0}\|^2 A + \lambda^2 \log(1/\delta_1) \sum_{k=1}^{s_t-1} \sigma_{\chi_{t,k}}^2 + A_\sigma \sum_{k=1}^{s_t-1} \|y_{t,k} - y_{t,k-1}\|,$$

where we define $A := 1 - \frac{2\lambda\mu L}{\mu+L} + \frac{\lambda^2 L^2 \log(1/\delta_1)}{S_2}$.

From Lemma 12 in (Luo et al. (2020)) we know

$$\left\|\frac{y_{i,1} - y_{i,0}}{\lambda}\right\|^2 \leqslant 3\|u_{i,0} - \nabla_y f(x_i, y_i)\|^2 + 3L^2\|x_i - x_{i-1}\|^2 + 3\|\gamma_{i-1}\|^2 \tag{35}$$
$$\leqslant 9\|\theta_{i-1}\|^2 + 21L^2\|x_i - x_{i-1}\|^2 + 3\|\gamma_{i-1}\|^2$$

Therefore,

$$\sum_{i=\lfloor t/q\rfloor q}^{t} \sum_{k=1}^{s_i} \|y_{i+1,k} - y_{i+1,k-1}\|^2 \leqslant \frac{\lambda^2 A}{1-A} \sum_{i=\lfloor t/q\rfloor q}^{t} \left\|\frac{y_{i+1,1} - y_{i+1,0}}{\lambda}\right\|^2 + \sum_{i=\lfloor t/q\rfloor q}^{t} \frac{\chi}{1-A}$$
$$\leqslant \frac{\lambda^2 A}{1-A} \sum_{i=\lfloor t/q\rfloor q}^{t} \left(3\|\theta_i\|^2 + 7L^2\|x_{i+1} - x_i\|^2 + \|r_i\|^2\right) + \sum_{i=\lfloor t/q\rfloor q}^{t} \frac{\chi}{1-A} + A_\sigma \sum_{i=t_0}^{t} \sum_{k=1}^{s_i-1} \|y_{i,k} - y_{i,k-1}\|,$$

where we denote $\chi := s_{i-1}\chi_{i,k}^2 \lambda^2$.

Using the choice of $\lambda \leqslant \frac{1}{6L}$ we can further conclude

$$\|\alpha_{t+1}\|^2 \leqslant 4\log(4/\delta_1)\left(\frac{\sigma^2}{S_1} + \sigma_{\omega_{\lfloor t/q\rfloor q}}^2 + \frac{4L^2\lambda^2 A}{(1-A)S_2} \sum_{i=\lfloor t/q\rfloor q}^{t} \left(8L^2\|x_{i+1} - x_i\|^2 + 3\|\theta_i\|^2 + \|\gamma_i\|^2\right)\right)$$
$$+ 8\log(1/\delta_1)qK\sigma_{\xi_{t,k}}^2 + A_\sigma \sum_{m=t_0}^{t} \sum_{k=1}^{s_m-1} \|y_{m,k} - y_{m,k-1}\|^2 \tag{36}$$

$$\|\theta_{t+1}\|^2 \leqslant 4\log(4/\delta_1)\left(\frac{\sigma^2}{S_1} + \sigma_{\tau_{\lfloor t/q\rfloor q}}^2 + \frac{4L^2\lambda^2 A}{(1-A)S_2} \sum_{i=\lfloor t/q\rfloor q}^{t} \left(8L^2\|x_{i+1} - x_i\|^2 + 3\|\theta_i\|^2 + \|\gamma_i\|^2\right)\right)$$
$$+ \log(1/\delta_1)qK\sigma_{\chi_{t,k}}^2 + A_\sigma \sum_{m=t_0}^{t} \sum_{k=1}^{s_m-1} \|y_{m,k} - y_{m,k-1}\|^2 \tag{37}$$

Next we will estimate the bound of $\|\gamma_i\|$. Define

$$\tilde{y}_{t,k+1} = \Pi_{\mathcal{Y}}(y_{t,k} + \lambda\nabla_y f(x_t, y_{t,k})) \tag{38}$$

Then according to the proof of SREDA (Lemma 10 Eq. (9) in (Luo et al., 2020)), we have

$$
\begin{aligned}
f(x_t, y_{t,k}) \leqslant{}& f(x_t, \tilde{y}_{t,k+1}) - \left(\frac{1}{2\lambda} - \frac{L}{2}\right) \|\tilde{y}_{t,k+1} - y_{t,k}\|^2 - \left(\frac{1}{3\lambda} - L\right) \|y_{t,k+1} - y_{t,k}\|^2 \\
&+ \lambda \|u_{t,k} - \nabla_y f(x_t, y_{t,k})\|^2 \\
\leqslant{}& f(x_t, \tilde{y}_{t,k+1}) - \left(\frac{1}{2\lambda} - \frac{L}{2}\right) \|\tilde{y}_{t,k+1} - y_{t,k}\|^2 - \left(\frac{1}{3\lambda} - L\right) \|y_{t,k+1} - y_{t,k}\|^2 \\
&+ 4\lambda \log(4/\delta_1) \left( \|u_{t,0} - \nabla_y f(x_t, y_{t,0})\|^2 + \frac{L^2}{S_2} \sum_{i=0}^{k-1} \|y_{t,i+1} - y_{t,i}\|^2 + k\sigma_{\chi_t}^2 \right)
\end{aligned}
\tag{39}
$$

where in the second inequality the property of sub-Gaussian is applied to $\|u_{t,k} - \nabla_y f(x_t, y_{t,k})\|^2$ which is similar to Eq. 34 to get

$$
\|u_{t,k} - \nabla_y f(x_t, y_{t,k})\|^2 \leqslant 4 \log(4/\delta_1) \left( \|u_{t,0} - \nabla_y f(x_t, y_{t,0})\|^2 + \frac{L^2}{S_2} \sum_{i=0}^{k-1} \|y_{t,i+1} - y_{t,i}\|^2 + \sum_{k=1}^{K} \sigma_{\chi_{t,k}}^2 \right)
\tag{40}
$$

Applying recursion on Eq. 39, for any $k \leqslant K$ we have

$$
\begin{aligned}
f(x_t, y_{t,1}) \leqslant{}& f(x_t, y_{t,k}) - \sum_{i=1}^{k} \left( \frac{1}{2\lambda} - \frac{L}{2} - \frac{4L^2 \lambda \log(4/\delta_1)}{S_2} \right) \|y_{t,i+1} - y_{t,i}\|^2 \\
&+ 4k\lambda \log(4/\delta_1) \left( \|u_{t,0} - \nabla_y f(x_t, y_{t,0})\|^2 + \frac{L^2}{S_2} \|y_{t,1} - y_{t,0}\|^2 \right) \\
\leqslant{}& f(x_t, y_{t,k}) - 2L^2 \sum_{i=1}^{k} \|y_{t,i+1} - y_{t,i}\|^2 - L\lambda^2 \sum_{i=1}^{k} \|G_\lambda(x_t, y_{t,i})\|^2 \\
&+ 4k\lambda \log(4/\delta_1) \left( \|u_{t,0} - \nabla_y f(x_t, y_{t,0})\|^2 + \frac{L^2}{S_2} \|y_{t,1} - y_{t,0}\|^2 + \sigma_{\chi_t}^2 \right)
\end{aligned}
$$

where we have used $\lambda \leqslant \frac{1}{6L}$ and the definition of $G_\lambda(x, y)$. Let $k = K$ we achieve

$$
\begin{aligned}
\sum_{k=1}^{K} \|G_\lambda(x_t, y_{t,k})\|^2 \leqslant{}& \frac{f(x_t, y^*(x_t)) - f(x_t, y_{t,1})}{L\lambda^2} - \frac{2L}{\lambda^2} \sum_{k=1}^{K} \|y_{t,k+1} - y_{t,k}\|^2 \\
&+ \frac{4K \log(1/\delta_1)}{L\lambda} (\|u_{t,0} - \nabla_y f(x_t, y_{t,0})\|^2 + \frac{L^2}{S_2} \|y_{t,1} - y_{t,0}\|^2) + \frac{\sqrt{2} \log(1/\delta_1)}{L\lambda} \sum_{k=1}^{K} \sum_{i=1}^{k-1} \sigma_{\chi_{t,k}}^2
\end{aligned}
$$

Due to the definition of $\tilde{G}_\lambda(y_{t,k})$, we have

$$
\begin{aligned}
\left\| \tilde{G}_\lambda(y_{t,k}) - G_\lambda(x_t, y_{t,k}) \right\| &= \frac{1}{\lambda^2} \|\Pi_{\mathcal{Y}}(y_{t,k} + \lambda u_{t,k}) - \Pi_{\mathcal{Y}}(y_{t,k} + \lambda \nabla_y f(x_t, y_{t,k}))\|^2 \\
&\leqslant \|u_{t,k} - \nabla_y f(x_t, y_{t,k})\|^2
\end{aligned}
$$

because of the non-expansion property of projection. Recall the selection of $s_t$. Then by Cauchy-Schwarz inequality, Eq. 40

and $\lambda = \frac{1}{6L}$ we have

$$
\begin{aligned}
&\|G_\lambda(x_t, y_{t,s_t})\|^2 \\
&\leqslant 2\|\tilde{G}_\lambda(y_{t,s_t})\|^2 + 2\|u_{t,s_t} - \nabla_y f(x_t, y_{t,s_t})\|^2 \\
&\leqslant \frac{2}{K} \sum_{k=1}^K \|\tilde{G}_\lambda(y_{t,k})\|^2 + 2\|u_{t,s_t} - \nabla_y f(x_t, y_{t,s_t})\|^2 \\
&\leqslant \frac{4}{K} \sum_{k=1}^K \left(\|G_\lambda(x_t, y_{t,k})\|^2 + \|u_{t,k} - \nabla_y f(x_t, y_{t,k})\|^2\right) + 2\|u_{t,s_t} - \nabla_y f(x_t, y_{t,s_t})\|^2 \\
&\leqslant \frac{4}{K} \sum_{k=1}^K \|G_\lambda(x_t, y_{t,k})\|^2 + \frac{16 \log(4/\delta_1)}{K} \sum_{k=1}^K \left(\|u_{t,0} - \nabla_y f(x_t, y_{t,0})\|^2 + \frac{L^2}{S_2}\|y_{t,i+1} - y_{t,i}\|^2 + \sum_{i=1}^{K-1} \sigma_{\chi_{t,k}}^2\right) \\
&\quad + 8 \log(4/\delta_1) \left(\|u_{t,0} - \nabla_y f(x_t, y_{t,0})\|^2 + \frac{L^2}{S_2} \sum_{i=0}^{s_t-1} \|y_{t,i+1} - y_{t,i}\|^2 + \sum_{i=1}^{s_t-1} \sigma_{\chi_{t,k}}^2\right) \\
&\leqslant \frac{4}{K} \left[\frac{F_0}{\lambda^2} - \frac{2L}{\lambda^2} \sum_{k=1}^K \|y_{t,k+1} - y_{t,k}\|^2 + \frac{4K \log(1/\delta_1)}{L\lambda}(\|u_{t,0} - \nabla_y f(x_t, y_t, 0)\|^2 + \frac{L^2}{S_2}\|y_{t,1} - y_{t,0}\|^2) + \frac{4 \log(1/\delta_1)}{L\lambda} \sum_{i=1}^K \sigma_{\chi_{t,k}}^2\right] \\
&\quad + \frac{16 \log(4/\delta_1)}{K} \sum_{k=1}^K \left(\|u_{t,0} - \nabla_y f(x_t, y_{t,0})\|^2 + \frac{L^2}{S_2}\|y_{t,i+1} - y_{t,i}\|^2 + \sum_{i=1}^{K-1} \sigma_{\chi_{t,k}}^2\right) \\
&\quad + 8 \log(4/\delta_1) \left(\|u_{t,0} - \nabla_y f(x_t, y_{t,0})\|^2 + \frac{L^2}{S_2} \sum_{i=0}^{s_t-1} \|y_{t,i+1} - y_{t,i}\|^2 + \sum_{i=1}^{s_t-1} \sigma_{\chi_{t,k}}^2\right) \\
&\leqslant \frac{144\kappa}{K}\|G_\lambda(x_t, y_{t,0})\|^2 + \left(\frac{144\kappa}{K} + 120 \log(4/\delta_1)\right) \|u_{t,0} - \nabla_y f(x_t, y_{t,0})\|^2 \\
&\quad + \frac{120 \log(4/\delta_1) L^2}{S_2}\|y_{t,1} - y_{t,0}\|^2 + 16 \log(1/\delta_1) \sum_{i=1}^{K-1} \sigma_{\chi_{t,k}}^2
\end{aligned}
\tag{41}
$$

(We need to note that actually $\sigma_{\chi_{t,k}}^2$ has nothing to do with index $k$) According to Lemma 8 in (Luo et al. (2020)) and Cauchy-Schwarz inequality we have

$$
\|G_\lambda(x_t, y_{t,0})\|^2 \leqslant 2L^2\|x_t - x_{t-1}\|^2 + 2\|\gamma_{t-1}\|^2
\tag{42}
$$

Therefore, combining Eq. 35, Eq. 41 and Eq. 42, for $\forall t$ we can conclude

$$
\begin{aligned}
\|\gamma_{t+1}\|^2 &\leqslant \left(\frac{288\kappa}{K} + \frac{10 \log(4/\delta_1)}{S_2}\right) \|\gamma_t\|^2 + \left(\frac{432\kappa}{K} + 390 \log(4/\delta_1)\right) \|\theta_t\|^2 + \left(\frac{1152\kappa}{K} + 750 \log(4/\delta_1)\right) L^2\|x_{t+1} - x_t\|^2 \\
&\quad + C_\sigma \log(1/\delta_1) \left(\frac{4}{K} \sum_{i=1}^K 24\sigma_{\chi_{t,k}}^2 + \sum_{k=1}^{s_t-1} \sigma_{\chi_{t,k}}^2 + \frac{4}{K} \sum_{i=1}^{K-1} 16\sigma_{\chi_{t,k}}^2\right)
\end{aligned}
\tag{43}
$$

Before diving into the details of convergence analysis, we give a thorough analysis on the parameter setting. Quantified by the number of total samples, we have

$$
\frac{T}{q} \cdot S_1 + T \cdot K \cdot S_2 \leqslant n.
$$

Therefore, we set $T = \mathcal{O}((\frac{n\varepsilon}{\sqrt{d \log(1/\delta)}})^{2/3})$, $S_1 = \mathcal{O}(n^{2/3})$, $q = S_2 = \mathcal{O}(((\frac{n\varepsilon}{\sqrt{d \log(1/\delta)}})^{1/3}))$

Applying union bound, with probability at least $1 - 4\delta_1$, Eq. 36, Eq. 37 and Eq. 43 hold for $\forall t$. In the descent phase we have $\|x_{t+1} - x_t\|^2 \leqslant \eta^2$. At the perturbation step we have $\|x_{t+1} - x_t\|^2 \leqslant r^2$. In the escaping phase, on average we have

$\|x_{t+1} - x_t\|^2 \leqslant D$. Thus, we have

$$\|x_{t+1} - x_t\|^2 \leqslant \max\{\eta^2, r^2, D\} \leqslant \frac{\sqrt{d \log(1/\delta)}}{\log^2(4/\delta_1)\kappa^2 L^2 C_\Phi^2 n\varepsilon} \tag{44}$$

According to the choices that $q = \mathcal{O}((\frac{n\varepsilon}{\sqrt{d \log(1/\delta)}})^{1/3})$, $K = O(\kappa)$, $S_1 = \mathcal{O}(n^{2/3})$ and $S_2 \geqslant \log^2(4/\delta_1)\kappa(\frac{n\varepsilon}{\sqrt{d \log(1/\delta)}})^{1/3}$, by induction we can prove for $\forall t$, the following bounds hold

$$\|\alpha_t\|^2 \leqslant O(\frac{1}{n^{2/3}} + (\frac{d \log(1/\delta_1)}{n^2\epsilon^2})^{1/2}) \tag{45}$$

$$\|\theta_t\|^2 \leqslant O(\frac{1}{n^{2/3}} + (\frac{d \log(1/\delta_1)}{n^2\epsilon^2})^{1/2}) \tag{46}$$

$$\|\gamma_t\|^2 \leqslant O(\frac{1}{n^{2/3}} + (\frac{d \log(1/\delta_1)}{n^2\epsilon^2})^{1/2}) \tag{47}$$

where the case of $t = 0$ is satisfied by the choice of $S_1$ and the PiSARAH initialization $\|\gamma_0\| \leqslant 1/\kappa(\frac{\sqrt{d \log(1/\delta)}}{n\varepsilon})^{1/2}$. By choosing $K = \mathcal{O}(\kappa)$, $D = \frac{\sqrt{d \log(1/\delta)}}{n\varepsilon}$ and take $d = \max\{d_1, d_2\}$

$$\begin{aligned}
\|v_t - \nabla\Phi(x_t)\| &= \|v_t - \nabla_x f(x_t, y^*(x_t))\| \\
&= \underbrace{\|v_t - \nabla_x f(x_t, y_{t+1})\|}_{=:\alpha_t} + \|(\nabla_x f(x_t, y_{t+1}) - \nabla_x f(x_t, y^*(x_t)))\| \\
&\leqslant \|\alpha_t\| + L \|y_{t+1} - y^*(x_t)\| \\
&\leqslant \|\alpha_t\| + 2\kappa \|\gamma_t\|
\end{aligned} \tag{48}$$

where the last inequality holds by Lemma 12 and we can further obtain $\|v_t - \nabla\Phi(x)\| \leqslant \|\alpha_t\| + 2\kappa\|\gamma_t\| \leqslant O(\frac{1}{n^{1/3}} + (\frac{\sqrt{d \log(1/\delta)}}{n\varepsilon})^{1/2})$.

$\square$

Next we will show the result of the decrease of loss function value $\Phi(x)$ in the descent phase.

**Lemma 9.** *Given the parameter setting as follows, stepsize $\eta_H$, escaping threshold $t_{thres} = 2\log(\frac{\eta_H \alpha_H L_\Phi}{C\rho_\Phi r_0})/\eta_H = \tilde{O}(\frac{1}{\eta_H \alpha_H})$, perturbation radius $r \leqslant R$ and average movement $\bar{D} \leqslant R^2/t_{thres}^2$ and $\alpha_H = \sqrt{\rho_\Phi \alpha}$. Then for $\forall s$, if our algorithm does not break the escaping phase, then we have $\lambda_{\min} \geqslant -\alpha_H$ with probability $1 - \delta_1 - \delta_2$.*

*Proof.* Let $\{x_t\}, \{x'_t\}$ be two coupled sequences by running 2 algorithm from $x_{m_s+1} = x_{m_s} + \xi$ and $x'_{m_s+1} = x'_{m_s} + \xi'$ with $x_{m_s+1} - x'_{m_s+1} = r_0 e_1$, where $\xi, \xi' \in B_0(r)$, $r_0 = \frac{\delta_2 r}{\sqrt{d}}$ and $e_1$ denotes the smallest eigenvector direction of $\nabla^2\Phi(x_{m_s})$. When $\lambda_{\min}(\nabla^2\Phi(x_{m_s})) \leqslant -(\frac{\sqrt{d \log(1/\delta)}}{n\varepsilon})$, by Lemma 10 we have

$$\max_{m_s < t \leqslant m_s + t_{thres}} \{\|x_t - x_{m_s}\|, \|x'_t - x_{m_s}\|\} \geqslant R$$

with probability at least $1 - 4\delta_1$.

Let $\mathcal{S}$ be the set of $x_{m_s+1}$ that will not generate a sequence moving out of the ball with center $x_{m_s}$ and radius $r$. Then the projection of $\mathcal{S}$ onto direction $e_1$ should not be larger than $r_0$. By integration, we can calculate the volume of the ball and stuck region in $d$-dimension and further check that the probability of $x_{m_s+1} \in \mathcal{S}$ is smaller than $\delta_2$ as $\xi$ is drawn from uniform distribution, which is shown in Eq. (49):

$$Pr(x_{m_s+1} \in \mathcal{S}) \leqslant \frac{r_0 V_{d-1}(r)}{V_d(r)} \leqslant \frac{\sqrt{d} r_0}{r} \leqslant \delta_2 \tag{49}$$

where $V_d(r)$ is the volume of $d$-dimension ball with radius $r$. Applying union bound, with probability at least $1 - 4\delta_1 - \delta_2$ we have

$$\exists m_s < t \leqslant m_s + t_{thres}, \quad \|x_t - x_{m_s+1}\| \geqslant R. \tag{50}$$

If Algorithm 2 does not break the escaping phase, then for $\forall m_s < t \leqslant m_s + t_{\text{thres}}$ we have

$$\|x_t - x_{m_s+1}\| < \sqrt{(t-m_s) \sum_{i=m_s+1}^{t-1} \|x_{i+1} - x_i\|^2} \leqslant (t-m_s)\sqrt{\bar{D}} \tag{51}$$

which is derived by Cauchy-Schwartz inequality. By the choice of parameters $t_{\text{thres}}$ and $\bar{D}$, we have

$$\|x_t - x_{m_s+1}\| < t_{\text{thres}}\sqrt{\bar{D}} \leqslant R. \tag{52}$$

Therefore, when $\lambda_{\min}(\nabla^2\Phi(x_{m_s})) \leqslant -\alpha_H$, with probability at least $1 - 4\delta_1 - \delta_2$ our Algorithm 2 will break the escaping phase. $\qquad \square$

**Lemma 10.** *Set stepsize $\eta_H \leqslant \frac{r_0}{2\|v_t\|} \leqslant \frac{1}{2}(\frac{\sqrt{d}}{n\varepsilon})^{-1}r_0$, and $R = \frac{1}{2L_\Phi\eta_H} = \frac{1}{L_\Phi r_0}(\frac{\sqrt{d}}{n\varepsilon})$, perturbation radius $r \leqslant \frac{L_\Phi\eta_H\alpha_H}{C\rho_\Phi}$ and threshold $t_{\text{thres}} = 2\log(\frac{\eta_H\alpha_H L_\Phi}{C\rho_\Phi r_0})/\eta_H = \tilde{O}(\frac{1}{\eta_H\alpha_H})$, where $r_0 \leqslant r$ and $C = \tilde{O}(1)$. Suppose $-\lambda_{\min}(\nabla^2\Phi(x_{m_s})) \leqslant -\alpha_H$.*

*Let $\{x_t\}, \{x'_t\}$ be two coupled sequences by running Algorithm 2 from $x_{m_s+1} = x_{m_s} + \xi$ and $x'_{m_s+1} = x_{m_s} + \xi'$ with $x_{m_s+1} - x'_{m_s+1} = r_0\mathbf{e}_1$, where $\xi, \xi' \in B_0(r)$ and $\mathbf{e}_1$ denotes the smallest eigenvector direction of $\nabla^2\Phi(x_{m_s})$. Then with probability at least $1 - 4\delta_1$ (for $\delta_1$ in Lemma 8), we have*

$$\max_{m_s < t \leqslant m_s + t_{thres}} \{\|x_t - x_{m_s}\|, \|x'_t - x_{m_s}\|\} \geqslant R. \tag{53}$$

*Proof.* To prove this lemma, we assume the contrary:

$$\forall m_s < t \leqslant m_s + t_{\text{thres}}, \quad \|x_t - x_{m_s}\| < R, \quad \|x'_t - x_{m_s}\| < R. \tag{54}$$

Define $w_t = x_t - x'_t$ and $\nu_t = v_t - \nabla\Phi(x_t) - (v'_t - \nabla\Phi(x'_t))$. We have

$$\begin{aligned} w_{t+1} &= w_t - \eta_H(v_t - v'_t) = w_t - \eta_H(\nabla\Phi(x_t) - \nabla\Phi(x'_t)) - \eta_H\nu_t \\ &= (I - \eta_H\mathcal{H})w_t - \eta_H(\Delta_t w_t + \nu_t) \end{aligned} \tag{55}$$

where

$$\mathcal{H} = \nabla^2\Phi(x_{m_s}), \qquad \Delta_t = \int_0^1 \left[\nabla^2\Phi(x'_t + \theta(x_t - x'_t)) - \mathcal{H}\right] d\theta. \tag{56}$$

Let

$$p_{t+1} = (I - \eta_H\mathcal{H})^{t-m_s}w_{m_s+1}, \qquad q_{t+1} = \eta_H \sum_{\tau=m_s+1}^{t} (I - \eta_H\mathcal{H})^{t-\tau}(\Delta_\tau w_\tau + \nu_\tau) \tag{57}$$

and apply recursion to Eq. (55), we can obtain

$$w_{t+1} = p_{t+1} - q_{t+1}. \tag{58}$$

Next, we will inductively prove

$$\|q_t\| \leqslant \|p_t\|/2, \quad \forall m_s < t \leqslant m_s + t_{\text{thres}}. \tag{59}$$

First, when $t = m_s + 1$ the conclusion holds since $\|q_{m_s+1}\| = 0$. Suppose the above equation is satisfied for $\tau \leqslant t$. Then we have

$$\|w_\tau\| \leqslant \|p_\tau\| + \|q_\tau\| \leqslant \tfrac{3}{2}\|p_\tau\| = \tfrac{3}{2}(1 + \eta_H\gamma)^{\tau-m_s-1}r_0. \tag{60}$$

Then for the case $\tau = t + 1$, by the above two equations we have

$$\|q_{t+1}\| \leqslant \eta_H (1 + \eta_H \gamma)^{t-m_s} \cdot \frac{3}{2} \sum_{\tau=m_s+1}^{t} \|\Delta_\tau\| r_0 + \eta_H \sum_{\tau=m_s+1}^{t} (1 + \eta_H \gamma)^{t-\tau} \|\nu_\tau\|$$

$$\leqslant (1 + \eta_H \gamma)^{t-m_s} \left( \eta_H L_\Phi R r_0 + \tfrac{1}{4} r_0 \right) \tag{61}$$

$$\leqslant \tfrac{1}{2} (1 + \eta_H \gamma)^{t-m_s} r_0 = \|p_{t+1}\|/2. \tag{62}$$

to get the above result, all we need is to assume that $L_\Phi \eta_H R \leqslant \frac{3}{4}$.

In the second inequality, we use Lipschitz Hessian to obtain $\|\Delta_\tau\| \leqslant L_\Phi R$ and we use Lemma 8 and the fact

$$a^{t+1} - 1 = (a - 1) \sum_{s=0}^{t} a^s$$

to obtain $\|\nu_\tau\| \leqslant \frac{\sqrt{d \log(1/\delta)}}{n\varepsilon}$ with probability $1 - 4\delta_1$. The last inequality can be achieved by the definitions of $\eta_H$ and $t_{\text{thres}}$.

Now we have

$$\tfrac{1}{2}(1 + \eta_H \gamma)^{t-m_s-1} r_0 \leqslant \|w_t\| \leqslant \|x_t - x_{m_s}\| + \|x_t' - x_{m_s}\| \tag{63}$$

which conflicts with Eq. (54) due to the choice of $t_{\text{thres}}$.

Note that $e_1$ is the eigenvector of Hessian $H$, i.e. $He_1 = -re_1$. Therefore, $P_{t+1} = (I - \eta_H H)^{t-m_s}$ $\qquad \square$

## C. Auxiliary Lemmas

**Lemma 11.** *Suppose $f$ is a $\mu$-strongly convex function and has L-Lipschitz gradient. Then for any $x$ and $x'$ we have*

$$\langle \nabla f(x) - \nabla f(x'), x - x' \rangle \geqslant \frac{\mu L}{\mu + L} \|x - x'\|^2 + \frac{1}{\mu + L} \|\nabla f(x) - \nabla f(x')\|^2.$$

**Lemma 12.** *For any $y \in \mathcal{Y}$ we have*

$$\frac{\mu}{2} \|y - y^*(x_t)\| \leqslant \|\mathcal{G}_\lambda(x_t, y)\|.$$

## D. Experimental Details

**Differential privacy.** Each data record $(A_i, b_i)$ is treated as sensitive. For each private gradient access, we clip per-sample gradients to $\ell_2$ norm at most $C = 1$, average over a mini-batch, and add Gaussian noise calibrated to sensitivity $2C/B$ (where $B$ is the mini-batch size). We fix the overall privacy budget to $(\varepsilon, \delta) = (2, 10^{-6})$ and allocate per-query noise according to the total number of oracle calls made by each method (using advanced composition in the code). This allocation is necessary because DP-RGDA consumes more private gradient calls due to inner updates and variance-reduction updates.

**Hyperparameters.** We run for $T = 400$ outer iterations with inner loop length $K = 5$ and SPIDER refresh period $q_{\text{period}} = 10$. We use batch sizes $b_1 = 200$ (refresh) and $b_2 = 50$ (incremental updates). Step sizes are: DP-RGDA $(\eta_x, \eta_y) = (0.2, 0.8)$, Sto-SPIDER $\eta_x = 0.005$, and Ada-DP-SPIDER $\eta = 0.02$. For the curvature diagnostic, we estimate $\lambda_{\min}$ at every iteration using a finite-difference Hessian–vector product with step size $h = 5 \times 10^{-4}$ and an iterative eigensolver (maxiter $500$, tolerance $10^{-4}$).

