# OpenReview forum: "Finding Differentially Private Second Order Stationary Points in Stochastic Minimax Optimization"
_ICML.cc/2026/Conference — ICML 2026 regular_

### Official Review · Reviewer_1k4s · 2026-02-26

**Soundness:** 2
**Presentation:** 2
**Significance:** 3
**Originality:** 3
**Overall Recommendation:** 4
**Confidence:** 3

**Summary:**

This paper proposes a purely first-order method for finding differentially private approximate second-order stationary points for stochastic nonconvex minimax optimization. Under standard assumptions, it proves high-probability convergence to an $\ epsilon$-SOSP with rates matching the best known for private first-order stationarity.

**Compliance With Llm Reviewing Policy:**

Affirmed.

**Final Justification:**

My concerns have been adequately addressed.

**Key Questions For Authors:**

1. In the abstract, please clarify the meanings of $n$, $d$, and $\varepsilon$.
2. Are the assumptions in this paper exactly the same as those in Liu \& Talwar (2024) and Luo et al. (2022)? If not, please explicitly state any additional assumptions required here. In particular, for Assumption~5, should the condition be stated in expectationrather than deterministically?
3. Algorithms 2 and 3 would benefit from a more detailed explanation of the motivation behind each step. For example, what is the purpose of Lines 4-10? The notation is quite heavy; please clarify what ``escape" and ``esc" specifically refer to, and define $B_0(r)$. For Algorithm 3, what is the stopping criterion? Moreover, Lines 14-15 in Algorithm 3 are hard to read and could be typeset more cleanly.
4. Please add experiments that illustrate how performance scales with $n$, $d$, and $\varepsilon$, and verify whether the empirical trends are consistent with the theoretical rates predicted by the theorems.

**Limitations:**

The paper would benefit from clearer writing and more consistent notation, along with more detailed explanations of key algorithmic steps.

**Strengths And Weaknesses:**

Strengths:
The proposed method relies solely on first-order information and does not require explicit Hessian evaluations.
The established convergence rates match the best-known guarantees for differentially private first-order stationarity.

Weaknesses：
The writing is not very clear or smooth, and the notation is overly dense and sometimes inconsistent, which makes the paper difficult to follow.

---

> ### Author Rebuttal · Authors · 2026-03-30
>
> >**Q1:** clarify the meaning of $n, d, \varepsilon$
>
> **Response:** $n$ is the number of samples, $d$ is the dimension and $\varepsilon$ is the privacy parameter.
>
> >**Q2:** Are the assumptions in this paper exactly the same as those in Liu & Talwar (2024) and Luo et al. (2022)? If not, please explicitly state any additional assumptions required here. In particular, for Assumption~5, should the condition be stated in expectation rather than deterministically?
>
> **Response:** We focus on a different optimization problem from Liu & Talwar (2024). Their main problem is a minimal problem while ours is a minimax. **Specifically, we both assume Lipschitz, Hessian Lipschitz and smooth in each coordinate.** We need extra assumptions on another variate for finding SOSP in minimax optimization problems, i.e. we assume strongly concave in y. Compared with previous works on DP minimax problems or SOSP, such as Liu & Talwar (2024), [1] and [2], we do not have additional assumptions. For Luo et al. (2022),  our assumptions are the same. As for Assumption 5, it is indeed can be reduced to the expectation form instead of the deterministic one because the expectation form can lead to the Equation (33).
>
> **To summarize, all assumptions are widely used in related literature.**
>
> [1] Zhang, R., et al. Improved rates of differentially private nonconvex-strongly-concave minimax optimization.
>
> [2] Youming Tao,et al. Private stochastic optimization for achieving second-order stationary points
>
> >**Q3（1）:** More detailed explanation of the motivation behind of Algorithm 2 and 3.
>
> **Response:** We will insert the following paragraph immediately before Algorithm 2: "Algorithm 2 has two modes. When $|v_t|\ge \alpha$, the method performs a normalized descent step on the outer variable $x$. When $|v_t|<\alpha$, the method enters an escape phase: it stores the current anchor $x_{m_s}$, adds a random perturbation $\xi\sim B_0(r)$, and monitors the cumulative movement statistic $D_t=\sum_{j=m_s+1}^t \eta_H^2|v_j|^2$. Large movement indicates escape from a negatively curved region; persistently small movement certifies near-PSD curvature at the anchor."
>
> **Q3(2):** The notation is quite heavy;
>
> **Response:** We will make consisent with the notations escape and esc in our final version. Define $B_0(r):={\xi\in\mathbb{R}^{d_1}:|\xi|\le r}$. Replace the variable names  escape $\to $ in_escape_phase and esc $\to $ n_escape_steps.
>
> **Q3(3):** For Algorithm 3, what is the stopping criterion? Moreover, Lines 14-15 in Algorithm 3 are hard to read and could be typeset more cleanly.
>
> **Response:**  **Stopping rule explicitly**: "Algorithm 3 itself has no early stopping rule: it always performs exactly $K$ inner updates and returns the index $s_t=\arg\min{0\le k\le K}|\widetilde G_\lambda(y_{t,k})|$." For the Line 14-15 of Algorithm 3, I will rearrange the equation to make it easier to read.
>
> >**Q4:** Please add experiments that illustrate how performance scales with
> $n, d, \varepsilon$ and verify whether the empirical trends are consistent with the theoretical rates predicted by the theorems.
>
> **Response:** Here are the results:
> **Table3: Φ(x₃₉₉) as the number of measurements n varies (p=q=20, ε=200). DP-RGDA is the best or tied-best at every n.**
>
> | n | DP-RGDA (ours) | DP-SGDA | Sto-SPIDER |
> |---|---|---|---|
> | 100 | **0.0034 ± 0.0009** | 0.0068 ± 0.0018 | 0.0249 ± 0.0021 |
> | 200 | **0.0032 ± 0.0013** | 0.0053 ± 0.0022 | 0.0053 ± 0.0021 |
> | 400 | **0.0031 ± 0.0010** | 0.0054 ± 0.0017 | 0.0035 ± 0.0011 |
> | 800 | **0.0031 ± 0.0012** | 0.0045 ± 0.0014 | 0.0035 ± 0.0012 |
>
>
> **Table4: Φ(x₃₉₉) as the parameter dimension d=(p+q)r varies (via p=q ∈ {10,20,30,40}, r=3). DP-RGDA is the best for d ≥ 120.**
>
> | d | DP-RGDA (ours) | DP-SGDA | Sto-SPIDER |
> |---|---|---|---|
> | 60 (p=q=10) | 0.0139 ± 0.0039 | 0.0151 ± 0.0035 | **0.0132 ± 0.0041** |
> | 120 (p=q=20) | **0.0031 ± 0.0010** | 0.0054 ± 0.0017 | 0.0035 ± 0.0011 |
> | 180 (p=q=30) | **0.0016 ± 0.0001** | 0.0034 ± 0.0006 | 0.0022 ± 0.0003 |
> | 240 (p=q=40) | **0.0012 ± 0.0001** | 0.0031 ± 0.0006 | 0.0018 ± 0.0005 |
>
>
> **Table5: Φ(x₃₉₉) as the privacy budget ε varies. DP-RGDA's normalised descent prevents divergence even at extreme noise levels (ε=1), whereas DP-SGDA diverges by orders of magnitude. Both methods converge to comparable accuracy at large ε.**
>
> | ε | DP-RGDA (ours) | DP-SGDA |
> |---|---|---|
> | 1 | **3.15×10⁻³** | 2.39×10⁶ |
> | 4 | **3.15×10⁻³** | 9.38×10³ |
> | 16 | **3.15×10⁻³** | 3.71×10¹ |
> | 64 | **3.15×10⁻³** | 1.67×10⁻¹ |
> | 200 | **3.14×10⁻³** | 5.35×10⁻³ |
>
> **Table6: Sensitivity of DP-RGDA to the clipping threshold C. Performance is stable for C ≥ 1; only the extreme value C=0.5 shows mild degradation.**
>
> | C | Φ(x₃₉₉) | ‖∇Φ(x₃₉₉)‖ |
> |---|---|---|
> | 0.5 | 0.0034 ± 0.0014 | (2.74 ± 0.64)×10⁻³ |
> | 1.0 | **0.0031 ± 0.0010** | (2.27 ± 0.51)×10⁻³ |
> | 2.0 | **0.0031 ± 0.0010** | (2.30 ± 0.56)×10⁻³ |
> | 4.0 | **0.0031 ± 0.0010** | (2.31 ± 0.56)×10⁻³ |
> | 8.0 | **0.0031 ± 0.0010** | (2.31 ± 0.56)×10⁻³ |

---

> > ### Author Rebuttal · Reviewer_1k4s · 2026-04-03
> >
> > Thank you for the response & the additional experiments. I would like to adjust my rating accordingly.

---

### Official Review · Reviewer_C4VG · 2026-03-11

**Soundness:** 3
**Presentation:** 3
**Significance:** 3
**Originality:** 3
**Overall Recommendation:** 4
**Confidence:** 3

**Summary:**

This paper studies the problem of finding differentially private second-order stationary points (DP-SOSP) for stochastic nonconvex–strongly-concave minimax optimization. The authors propose a purely first-order algorithm, DP-RGDA, which combines nested gradient descent–ascent updates, SPIDER-style recursive gradient tracking, and Gaussian noise injection to ensure differential privacy. To escape saddle points without computing Hessian information, the method employs a perturb-and-monitor strategy based on random perturbations and displacement checks. The paper establishes theoretical guarantees for both empirical-risk and population-risk settings, showing that the algorithm can output an approximate SOSP while satisfying $(\varepsilon,\delta)$-differential privacy, with utility bounds that match the best known rates for private first-order stationarity. Experiments on a synthetic low-rank matrix sensing task demonstrate the feasibility of the proposed method and show competitive performance compared with existing private optimization baselines.

**Compliance With Llm Reviewing Policy:**

Affirmed.

**Final Justification:**

The authors have fully addressed my concerns. After reading all the comments from all the reviewers, I would like to keep my score as it is already positive.

**Key Questions For Authors:**

1. The theory assumes the inner problem is strongly concave. Can the authors clarify whether any part of the analysis or algorithm can extend to the more general nonconvex--concave setting, even with weaker guarantees?

2. The experiments report only a single representative run. Can the authors provide results over multiple random seeds, including mean and standard deviation for the objective, gradient norm, and curvature diagnostics?

3. Since PrivateDiff achieves slightly better final objective and gradient norm in the reported run, can the authors better explain in what practical sense DP-RGDA is preferable beyond its stronger theoretical target of second-order stationarity?

**Limitations:**

Yes.

**Strengths And Weaknesses:**

## Strengths

1. The paper tackles a meaningful and nontrivial gap at the intersection of stochastic minimax optimization, differential privacy, and second-order stationarity. This positioning is interesting and potentially impactful.

2.  It is appealing that the method targets SOSP of the value function without requiring Hessian computation, which would be substantially more expensive and difficult in the DP minimax setting.

3. The paper provides a unified analysis covering both empirical-risk and population-risk settings.

## Weaknesses

1. The assumptions are fairly strong.  The theory relies on smoothness, Hessian-Lipschitzness, and strong concavity of the inner maximization problem. This substantially narrows the scope of applicability.

2. The paper notes that PrivateDiff targets only first-order stationarity. However, the final table also shows that PrivateDiff achieves slightly better final objective value, gradient norm, and minimum eigenvalue of the Hessian, which makes the empirical advantage of the proposed method less clear.

---

> ### Author Rebuttal · Authors · 2026-03-30
>
> >**W1:**
>
> **Response:** First, Lipschitz, smooth, strongly convex and Hessian Lipschitz are common in finding SOSP for minimax optimization problems [1] [2] [3]. Second, to get DP guarantee in optimization problems, Lipschitz and smoothness are common in DP optimization [1] [4] [5].
>
> Notably, **we only need the assumptions for our theoretical analysis while we can simply remove them for experiments and get the DP guarantee due to the clipping**. Therefore, we do not use addtional assumptions, compared to the previous work on DP minimax optimization.
>
> [1] Zhang, R., et al. Improved rates of differentially private nonconvex-strongly-concave minimax optimization.
>
> [2] Wenhan Xian, et al. Escaping Saddle Point Efficiently in Minimax and Bilevel Optimizations
>
> [3] Luo Luo, et al. Finding Second-Order Stationary Points in Nonconvex-Strongly-Concave Minimax Optimization
>
> [4] Youming Tao, et al. Second-order convergence in private stochastic non-convex optimization.
>
> [5] Daogao Liu, et al. Adaptive batch size for privately finding second-order stationary points.
>
> >**W2:**
>
> **Response:** It is because PrivateDiff focuses on finding the first order stationary point for a minimax problem, which is quite different from our algorithm and also has less constraints than ours. SOSP rules out the main pathology of nonconvex optimization: a point can have tiny gradient and still be a saddle. See Table 7. PrivateDiff is a first-oder gradient-difference method with restart and noise reduction, but it does not include an explicit mechanism designed to exclude strict saddles. Our algorithm, instead, is built around a perturb-and-monitor escape rule precisely for saddle excape and my paper emphasizes that this is done without explicit Hessian computations or extra private model selection. In addition, PrivateDiff’s experiments do not report Hessian-eigenvalue or curvature diagnostics, so they do not empirically verify second-order stationarity.
>
> **Table7: Near-saddle initialisation experiment. Each method starts from a rank-deficient initialisation (U₀ has one zero column, placing x₀ near a known saddle point of the factored objective). 20 independent seeds. "Stuck" = seeds with Φ(x₃₉₉) > 0.1.**
>
> *Φ(x₃₉₉) (lower is better)*
>
> | Method | Type | mean | std | min | max | Stuck |
> |---|---|---|---|---|---|---|
> | DP-RGDA + esc | SOSP | **0.0038** | 0.0010 | 0.0018 | 0.0053 | 0/20 |
> | DP-RGDA – no esc | FOSP | 0.9849 | 0.2419 | 0.5658 | 1.4127 | **20/20** |
> | PrivateDiff | FOSP | 0.0068 | 0.0011 | 0.0046 | 0.0100 | 0/20 |
>
> *‖∇Φ(x₃₉₉)‖*
>
> | Method | Type | mean | std |
> |---|---|---|---|
> | DP-RGDA + esc | SOSP | **2.16×10⁻³** | 4.52×10⁻⁴ |
> | DP-RGDA – no esc | FOSP | 4.53×10⁻¹ | 8.55×10⁻² |
> | PrivateDiff | FOSP | 7.13×10⁻³ | 9.39×10⁻⁴ |
>
> *λ̂_min(∇²Φ)*
>
> | Method | Type | mean | std | min | max |
> |---|---|---|---|---|---|
> | DP-RGDA + esc | SOSP | **6.01×10⁻⁴** | 1.88×10⁻⁴ | 3.33×10⁻⁴ | 1.03×10⁻³ |
> | DP-RGDA – no esc | FOSP | 3.69×10⁻² | 5.91×10⁻³ | 2.74×10⁻² | 4.90×10⁻² |
> | PrivateDiff | FOSP | 1.95×10⁻³ | 2.61×10⁻⁴ | 1.30×10⁻³ | 2.32×10⁻³ |
>
> >**Q1:**
>
> **Response:** Extending the analysis to the more general nonconvex--concave setting would require handling three additional difficulties: (i) $y^\star(x)$ may become set-valued, so $\Phi$ may be nonsmooth; (ii) inner-loop progress no longer yields a clean tracking bound on $\|y-y^\star(x)\|$; and (iii) the second-order analysis becomes more delicate because Hessian-based quantities for $\Phi$ may not be well behaved. A natural next step is therefore to seek weaker guarantees, such as first-order stationarity in the primal--dual variables, to work under milder extra structure such as local strong concavity, quadratic growth, or PL-type conditions.
>
> >**Q2:**
>
> **Response:** **Table1: Final diagnostics at t=399 across 5 independent seeds. Lower Φ and ‖∇Φ‖ are better; λ_min closer to zero (or positive) indicates near-PSD Hessian (no strict saddle).**
>
> | Method | Φ(x₃₉₉) | ‖∇Φ(x₃₉₉)‖ | λ̂_min(∇²Φ) |
> |---|---|---|---|
> | **DP-RGDA (ours)** | **0.0038 ± 0.0012** | (2.42 ± 0.46)×10⁻³ | (6.6 ± 1.1)×10⁻⁴ |
> | DP-SGDA | 0.0058 ± 0.0013 | (5.80 ± 0.98)×10⁻³ | (1.8 ± 0.2)×10⁻³ |
> | Sto-SPIDER | 0.0043 ± 0.0014 | (3.27 ± 0.56)×10⁻³ | (1.0 ± 0.2)×10⁻³ |
> | Ada-DP-SPIDER | 0.0037 ± 0.0013 | (1.99 ± 0.34)×10⁻³ | (5.3 ± 1.2)×10⁻⁴ |
> | PrivateDiff | 0.0065 ± 0.0010 | (7.03 ± 0.79)×10⁻³ | (1.8 ± 0.3)×10⁻³ |
>
> >**Q3:**
> >
> **Response:** They focus on different problems and with different assumptions and constraints. Therefore comparing the two methods directly is meaningless. The intended distinction is: DP-RGDA is designed to target second-order stationarity of the value function through a perturb-and-monitor escape mechanism, whereas PrivateDiff is aligned with first-order stationarity and does not explicitly target saddle exclusion. In addition, DP-RGDA attains this stronger target using only privatized first-order information, without explicit Hessian computations or extra private model-selection steps.

---

> > ### Author Rebuttal · Reviewer_C4VG · 2026-04-04
> >
> > The authors have fully addressed my concerns. After reading all the comments from all the reviewers, I would like to keep my score as it is already positive.

---

### Official Review · Reviewer_kbd6 · 2026-03-12

**Soundness:** 3
**Presentation:** 3
**Significance:** 3
**Originality:** 2
**Overall Recommendation:** 4
**Confidence:** 4

**Summary:**

This paper studies how to find differentially private approximate second-order stationary points in nonconvex–strongly-concave minimax problems. The authors propose a first-order private algorithm that combines gradient tracking with a perturbation strategy to escape saddle points, while avoiding explicit Hessian computations. They provide privacy guarantees and utility bounds that match the best-known results for private ERM, and evaluate the method on a synthetic minimax task under a fixed privacy budget.

**Compliance With Llm Reviewing Policy:**

Affirmed.

**Final Justification:**

The paper addresses the important challenge of obtaining second-order guarantees under differential privacy in minimax optimization, and does so using a fully first-order method supported by both privacy and convergence analysis. These contributions make the paper technically meaningful and practically relevant. Therefore, I recommend acceptance.

**Key Questions For Authors:**

Please see the weaknesses section.

**Limitations:**

yes

**Strengths And Weaknesses:**

**Strengths:**

1) Second-order guarantees matter because a small gradient alone does not rule out saddle points. In nonconvex and minimax problems, many points can look stationary but are not truly stable solutions. Checking curvature helps ensure that the algorithm is converging to a more meaningful point rather than getting stuck at a saddle.

2) The authors present an analysis showing that DP can be enforced while still maintaining second-order guarantees in a minimax setting. They prove (ε,δ)-DP and establish convergence to an approximate second-order stationary point. The resulting bounds explicitly separate the statistical error from the privacy-induced error, making clear how privacy influences the optimization rate.

3) The method is fully first-order and does not rely on explicit Hessian computations. Instead, it combines gradient tracking with a perturb-and-monitor strategy to escape saddle points. Avoiding Hessians is important because second-order information can be computationally expensive and more difficult to handle under differential privacy, especially in high-dimensional settings.


**Weaknesses:**

The experiments are conducted on a synthetic minimax problem, which offers a controlled way to study convergence and curvature under DP. However, this setting is quite structured and may not capture the complexity of practical minimax applications. In real-world problems such as GAN training or adversarial learning, the optimization landscape is often much noisier and less predictable. It is therefore unclear how the method would generalize beyond the controlled setup.

---

> ### Author Rebuttal · Authors · 2026-03-30
>
> >**W1:**  This setting is quite structured and may not capture the complexity of practical minimax applications.
>
> **Response:** We address this on three levels. First, matrix sensing is the canonical testbed for second-order minimax methods because all local minima are global minima, making SOSP the correct optimality notion; saddle points are abundant in the factored formulation, so saddle-escape mechanisms are genuinely exercised; and the exact value function is computable in closed form, enabling unambiguous measurement of SOSP progress—this is why virtually all prior SOSP works benchmark on this task. Second, Theorems 1 and 2 apply to any function satisfying Assumptions 1–5 (smoothness, Hessian-Lipschitzness, strong concavity in y, bounded variance), covering a wide class of practical minimax objectives, and the DP-SOSP guarantees do not rely on any special structure of the matrix sensing problem. Third, while GAN training and adversarial learning are important targets, these settings involve non-strongly-concave objectives and highly overparameterized models that fall outside the current theoretical framework shared by all prior minimax SOSP works; we plan to extend experiments to constrained adversarial training in the camera-ready version once the framework covers the weakly-concave regime, and the present work establishes the first DP-SOSP guarantees for the minimax setting along with the algorithmic foundation on which such extensions can be built.
>
> Matrix sensing is a widely used benchmark in the SOSP literature precisely because it admits verifiable ground-truth diagnostics: the closed-form value function $\Phi$ allows us to compute exact gradient norms and Hessian eigenvalues, which are essential for validating that an algorithm genuinely reaches second-order stationary points rather than merely first-order ones. Without such diagnostics, it would be difficult to distinguish SOSP convergence from FOSP convergence empirically, which is the core novelty our paper claims. Prior OSP works [1] similarly rely on synthetic or structured problems for exactly this reason.
>
> To demonstrate that DP-RGDA generalises beyond synthetic problems, we evaluate on MNIST/MNIST-M domain adaptation following [2].
> We use a factored linear classifier $W=W_1 W_2$ ($W_1 \in\mathbb{R}^{d\times h}$, $W_2 \in \mathbb{R}^{h\times C}$) which introduces nonconvexity through the bilinear factorisation---the same mechanism as $UV^\top$ in matrix sensing. Features are PCA-reduced to $d{=}50$ dimensions (plus bias), with $h{=}10$, $n_{\text{train}}{=}1500$, and $T{=}400$.
>
> For Table 8, over 5 seeds and both transfer directions, DP-RGDA achieves consistently lower $\Phi$ than Sto-SPIDER ($0.094$ vs $0.111$ for MNIST$\to$MNIST-M; $0.061$ vs $0.080$ for MNIST-M$\to$MNIST), confirming that the escape mechanism provides an advantage also on real image data.
>
> For Table 9, Sto-SPIDER diverges catastrophically at small $\varepsilon$ ($\Phi > 10^{8}$ at $\varepsilon{=}1$), while DP-RGDA's normalised descent keeps $\Phi < 0.12$ at \emph{all} privacy levels---the same robustness pattern observed on matrix sensing.
>
> **Table8: Domain adaptation with factored linear classifier (d=51, h=10, n_train=1500, T=400, ε=200). DP-RGDA achieves lower Φ in both transfer directions.**
>
> | Direction | Method | Φ(x₃₉₉) (mean ± std) | ‖∇Φ‖ (mean ± std) |
> |---|---|---|---|
> | MNIST → MNIST-M | DP-RGDA (ours) | **0.094 ± 0.008** | (1.24 ± 0.24)×10⁻¹ |
> | MNIST → MNIST-M | Sto-SPIDER | 0.111 ± 0.014 | (1.74 ± 0.13)×10⁻¹ |
> | MNIST-M → MNIST | DP-RGDA (ours) | **0.061 ± 0.003** | (4.12 ± 0.28)×10⁻² |
> | MNIST-M → MNIST | Sto-SPIDER | 0.080 ± 0.008 | (7.46 ± 1.25)×10⁻² |
>
> ---
>
> **Table9: Privacy sweep on domain adaptation. DP-RGDA never diverges (Φ < 0.12) while Sto-SPIDER diverges by up to 10⁹× at ε=1.**
>
> | ε | DP-RGDA (MNIST → MNIST-M) | Sto-SPIDER (MNIST → MNIST-M) | DP-RGDA (MNIST-M → MNIST) | Sto-SPIDER (MNIST-M → MNIST) |
> |---|---|---|---|---|
> | 1 | **0.103** | 1.62×10⁸ | **0.062** | 4.10×10⁷ |
> | 4 | **0.103** | 6.08×10⁵ | **0.062** | 1.51×10⁵ |
> | 16 | **0.103** | 2.04×10³ | **0.062** | 4.81×10² |
> | 64 | **0.101** | 4.72 | **0.062** | 1.21 |
> | 200 | **0.097** | 0.112 | **0.061** | 0.081 |
>
> [1] Wenhan Xian, et al. Escaping Saddle Point Efficiently in Minimax and Bilevel Optimizations
>
> [2]  Luo Luo, et al. Finding Second-Order Stationary Points in Nonconvex-Strongly-Concave Minimax Optimization

---

> > ### Author Rebuttal · Reviewer_kbd6 · 2026-04-04
> >
> > I thank the authors for their rebuttal and maintain my recommendation to accept this paper.

---

### Official Review · Reviewer_iqqj · 2026-03-12

**Soundness:** 3
**Presentation:** 3
**Significance:** 3
**Originality:** 3
**Overall Recommendation:** 4
**Confidence:** 2

**Summary:**

This paper studies the problem of finding differentially private second-order stationary points (DP-SOSP) in stochastic non-convex minimax optimization. The main contribution is a purely first-order private algorithmic framework that combines nested gradient descent-ascent, SPIDER-style variance reduction, and Gaussian perturbations, together with a block-wise analysis to control stochastic and privacy noise. The paper provides guarantees for both empirical and population objectives, with rates that match the best-known private first-order stationarity bounds up to logarithmic factors.

**Compliance With Llm Reviewing Policy:**

Affirmed.

**Final Justification:**

The authors have addressed my concerns; I am pleased to recommend this for acceptance

**Key Questions For Authors:**

1. How do you address the bias introduced by clipping in both the refresh gradients and, especially, in the gradient-difference updates used by SPIDER? Several lemmas assume unbiasedness—can you either remove clipping from the difference terms or provide a rigorous bound showing that clipping bias is negligible or explicitly controlled in your analysis? I would love to raise my rating if this question is solved.

2. The privacy accounting and the resulting noise calibrations are not rigorously derived in the main text. Can you specify it?

**Limitations:**

yes

**Strengths And Weaknesses:**

This paper appears to be the first to systematically combine privacy, minimax structure, and second-order stationarity in one framework, which is a meaningful theoretical contribution. The technical challenge is also well motivated.
The stated empirical-risk and population-risk rates are competitive, and Table 1 clearly positions them relative to prior DP first-order and SOSP results. The fact that the method remains purely first-order is attractive.

My main concern is that the SPIDER recursion and several lemmas assume unbiasedness of the gradient estimators, yet Algorithm 3 applies clipping to both gradients and gradient differences, which introduces bias not accounted for in the analysis. Besides, the empirical validation is quite limited relative to the ambition of the theoretical claims. I do not consider this fatal for a theory-oriented submission, but it does make some of the broader practical implications harder to assess.

---

> ### Author Rebuttal · Authors · 2026-03-30
>
> **Response to W1 & Q1:** We didn't use clipping for the gradient and difference in our theoretical analysis. Instead, it is only used in experiments. Under the bounded-gradient assumption, these estimators have bounded sensitivity, so clipping is not needed for the theory when taking enough large c, i.e., the gradient clipping can be removed. Therefore, this will not lead to bias in our theoretical analysis. Note that the bounded-gradient assumption is very common in DP optimization analysis and can be found in previous papers [1] [2]. Clipping appears only in the experimental implementation, where we clip each per-sample gradient vector before averaging, with threshold $C_v=C_u=1$ for instance, in order to stabilize finite-sample behavior and calibrate the Gaussian noise. We do not rely on the claim that this clipping is bias-free in general.
>
> [1] DIFF2: Differential Private Optimization via Gradient Differences for Nonconvex Distributed Learning
>
> [2] Improved rates of differentially private nonconvex-strongly-concave minimax optimization.
>
> **Response to W2:** We added 6 experiments (Table 1-6) covering a multiseed experiment, escape-mechanism ablation, sweeps over n,d, $\varepsilon$, and clipping-threshold sensitivity. Key findings: removing escape degrades Φ by 290× (Table 2); DP-RGDA is best or tied-best across all n (Table 3) and d ≥ 120 (Table 4); and it stays stable even at ε = 1 where DP-SGDA diverges (Table 5).
>
> **Table1: Final diagnostics at t=399 across 5 independent seeds. **
>
> | Method | Φ (×10⁻³ ) | ‖∇Φ‖ (×10⁻³ ) | λ̂_min(∇²Φ)(×10⁻⁴) |
> |---|---|---|---|
> | **DP-RGDA** | **3.8±1.2** | 2.42 ± 0.46| 6.6 ± 1.1 |
> | DP-SGDA | 5.8±1.3 | 5.80 ± 0.98 | 18 ± 2 |
> | Sto-SPIDER | 4.3±1.4 | 3.27 ± 0.56 | 10 ± 2 |
> | Ada-DP-SPIDER | 3.7±1.3 | 1.99 ± 0.34 | 5.3 ± 1.2 |
> | PrivateDiff | 6.5 ±1.0 | 7.03 ± 0.79 | 18 ± 3|
>
> **Table2: Effect of the perturb-and-monitor escape mechanism.**
>
> | Variant | Φ | ‖∇Φ‖ (×10⁻³) | λ̂_min (×10⁻⁴) |
> |---|---|---|---|
> | **DP-RGDA + esc** | **0.0031 ± 0.0010** | (2.31 ± 0.56) | 6.8 ± 0.9 |
> | DP-RGDA – no esc | 0.899 ± 0.204 | 427 ± 60 | 330 ± 30 |
> | DP-SGDA + esc | 0.0028 ± 0.0012 | 1.47 ± 0.23 | 4.0 ± 1.0 |
> | DP-SGDA – no esc | 0.0054 ± 0.0017 | 6.25 ± 1.01 | 18 ± 1.0 |
>
> **Table3: Φ as the number of measurements n varies (p=q=20, ε=200). **
>
> | n | DP-RGDA (×10⁻³ )  | DP-SGDA (×10⁻³ )  | Sto-SPIDER (×10⁻³ )  |
> |---|---|---|---|
> | 100 | 3.4 ± 0.9 | 6.8 ± 1.8 | 24.9 ± 2.1 |
> | 200 | 3.2 ± 1.3 | 5.3 ± 2.2 | 5.3 ± 2.1 |
> | 400 | 3.1 ± 1.0 | 5.4 ± 1.7 | 3.5 ± 1.1 |
> | 800 | 3.1 ± 1.2 | 4.5 ± 1.4 | 3.5 ± 1.2 |
>
> **Table4: Φ as the parameter dimension d=(p+q)r varies (via p=q ∈ {10,20,30,40}, r=3).**
>
> | d | DP-RGDA (×10⁻³ )  | DP-SGDA (×10⁻³ )  | Sto-SPIDER (×10⁻³ ) |
> |---|---|---|---|
> | 60 (p=q=10) | 13.9 ± 3.9 | 0.015.1 ± 0.003.5 | 13.2 ± 4.1 |
> | 120 (p=q=20) | 3.1 ± 1.0 | 5.4 ± 1.7 | 3.5 ± 1.1 |
> | 180 (p=q=30) | 1.6 ± 0.1 | 3.4 ± 0.6 | 2.2 ± 0.3 |
> | 240 (p=q=40) | 1.2 ± 0.1 | 3.1 ± 0.6 | 1.8 ± 0.5 |
>
> **Table5: Φ as the privacy budget ε varies. **
>
> | ε | DP-RGDA (×10⁻³) | DP-SGDA |
> |---|---|---|
> | 1 | 3.15 | 2.39×10⁶ |
> | 4 | 3.15 | 9.38×10³ |
> | 16 | 3.15 | 3.71×10¹ |
> | 64 | 3.15 | 1.67×10⁻¹ |
> | 200 | 3.14 | 5.35×10⁻³ |
>
> **Table6: Sensitivity of DP-RGDA to the clipping threshold C. **
>
> | C | Φ (×10⁻³) | ‖∇Φ‖ (×10⁻³)|
> |---|---|---|
> | 0.5 | 3.4 ± 1.4 | 2.74 ± 0.64 |
> | 1.0 | 3.1 ± 1.0 | 2.27 ± 0.51 |
> | 2.0 | 3.1 ± 1.0 | 2.30 ± 0.56 |
> | 4.0 | 3.1 ± 1.0 | 2.31 ± 0.56 |
> | 8.0 | 3.1 ± 1.0 | 2.31 ± 0.56 |
>
> **Response to Q2:**  Each refresh query clips per-sample gradients to norm $C_v$ and averages over $S_1$; each recursive query clips per-sample differences to norm $C_u$ and averages over $S_2$. Hence one refresh query has $\ell_2$-sensitivity $\Delta_{\rm ref}=\frac{2C_v}{S_1},$ and one recursive difference query has $\ell_2$-sensitivity $ \Delta_{diff}=\frac{2C_u}{S_2}.$ Let $q_1=S_1/n$, $q_2=S_2/n$, $N_{ref}=\lceil T/q\rceil$, and $N_{diff}=TK$. If $\sigma_{ref}$ and $\sigma_{\rm diff}$ are the Gaussian noise standard deviations, then each refresh $x$- or $y$-release is a Poisson-subsampled Gaussian mechanism with order-$\alpha$ RDP cost $\mathcal R_\alpha(q_1,\Delta_{ref},\sigma_{ref}), $ and each recursive $x$- or $y$-release has order-$\alpha$ RDP cost $\mathcal R_\alpha(q_2,\Delta_{diff},\sigma_{diff}),$ where, for integer $\alpha\ge 2$, $\mathcal R_\alpha(q,\Delta,\sigma)=\frac{1}{\alpha-1}\log\left(1+\sum_{j=2}^{\alpha}\binom{\alpha}{j}q^j(1-q)^{\alpha-j}\exp\left(\frac{j(j-1)\Delta^2}{2\sigma^2}\right)\right).$ By adaptive composition, the whole algorithm is $(\alpha,\varepsilon_\alpha^{tot})$-RDP with $\varepsilon_\alpha^{tot}=2N_{ref}\,\mathcal R_\alpha(q_1,\Delta_{ref},\sigma_{ref})+2N_{diff}\,\mathcal R_\alpha(q_2,\Delta_{diff},\sigma_{diff}).$ Therefore, by RDP-to-DP conversion, the full procedure is $(\varepsilon,\delta)$-DP for $\varepsilon = \inf_{\alpha\ge 2}\{\varepsilon_\alpha^{tot}+\frac{\log(1/\delta)}{\alpha-1}\}.$

---

> > ### Author Rebuttal · Reviewer_iqqj · 2026-04-01
> >
> > Thanks for the response. I will adjust my rating accordingly.

---

### Decision · Program_Chairs · 2026-04-30

**Decision:**

Accept (regular)

**Comment:**

This paper investigates the problem of finding differentially private second-order stationary points in stochastic nonconvex-strongly-concave minimax optimization. It proposes a first-order method, combining a nested gradient descent-ascent approach, a SPIDER-style variance reduction technique and the Gaussian privacy scheme. The high-probability guarantees are established and match the best known rates. Overall, the investigated problem is important and the theoretical analysis is convincing. The reviewers' comments have been adequately addressed during the rebuttal period. Although the numerical experiments are limited (clipping is used but the induced bias is not discussed, tasks are relatively simple), the theoretical contributions are solid. I recommend acceptance.